# NONMYOPIC BAYESIAN OPTIMIZATION IN DYNAMIC COST SETTINGS

## ABSTRACT

Bayesian optimization (BO) is a popular framework for optimizing black-box functions, leveraging probabilistic models such as Gaussian processes. However, conventional BO assumes static query costs, which limits its applicability to real-world problems with dynamic cost structures, such as geological surveys or biological sequence design, where query costs vary based on previous actions. To address this, we propose a cost-constrained nonmyopic BO algorithm that incorporates dynamic cost models. Our method employs a neural network policy for variational optimization over multi-step lookahead horizons to plan ahead in dynamic cost environments. Empirically, we benchmark our method on synthetic functions exhibiting a variety of dynamic cost structures. Furthermore, we apply our method to a real-world application in protein sequence design using a large language model-based policy, demonstrating its scalability and effectiveness in handling multi-step planning in a large and complex query space. Our nonmyopic BO algorithm consistently outperforms its myopic counterparts in both synthetic and real-world settings, achieving significant improvements in both efficiency and solution quality.

## 1 INTRODUCTION

Bayesian optimization (BO) (Kushner, 1962; 1964; Shahriari et al., 2016; Frazier, 2018; Garnett, 2022) is a powerful tool for optimizing black-box functions by employing a probabilistic surrogate model, typically a Gaussian process, together with an acquisition function, to balance exploration and exploitation of the unknown objective function. In conventional BO, query costs are typically assumed to be static. The assumption of static query costs can be an obstacle to applying BO in practical applications where query costs may vary dynamically on a per-iteration basis (Aglietti et al., 2021; Lee et al., 2021; Folch et al., 2022; 2024). For instance, in geological surveys, the cost of querying a location varies based on its proximity to the previous query due to transportation expenses (Bordas et al., 2020). Another example is biological sequence design, where editing one token at a time incurs a low cost, but moving beyond the edit distance of one token becomes prohibitively expensive (Belanger et al., 2019). These environments exhibit a dynamic cost structure, where the query cost at a given location might depend on the last query or even the entire query history. Incorporating these cost structures into the decision-making process can greatly improve the quality of the solution returned by BO algorithms. These cost structures dynamically constrain the effective input space where the decision-making algorithm can move, requiring the agent to plan its decision by looking multiple steps into the future.

Nonmyopic BO incorporates lookahead steps to make more informed decisions at the current timestep (González et al., 2016; Astudillo et al., 2021; Yue & Kontar, 2020; Jiang et al., 2020a). One potential approach to solve nonmyopic BO in a dynamic cost environment is to view it as a Markov Decision Process (MDP) (Garcia & Rachelson, 2013; Puterman, 2014). MDPs are commonly used to model sequential decision-making problems. In the context of nonmyopic BO in a dynamic cost environment, where we aim to determine the optimal next action in a sample-efficient manner, MDP frames the decision process as a cost-constrained model-based reinforcement learning (CMBRL) problem, where the queried inputs are states and actions influence the transitions between consecutive states. Traditional CMBRL approaches, which rely on world models to simulate the environment (Janner et al., 2019; Wang & Ba, 2020; Hafner et al., 2021; Hamed et al., 2024), are not directly applicable for nonmyopic BO settings.

On one hand, CMBRL is inadequate to handle various requirements in many nonmyopic BO applications under dynamic cost settings. Indeed, CMBRL algorithms often struggle with handling large, complex, and semantically rich action spaces as they implement policy with a simple feed-forward neural network that can typically handle a small, discrete action space (Janner et al., 2019; Wang & Ba, 2020; Hafner et al., 2021). In the above example of biological sequences, using a simple model is often inadequate if we want to incorporate domain knowledge during policy optimization. This is especially important when dealing with high-dimensional and complex action spaces, like editing sequences, where each action has rich semantic meaning and can significantly impact the outcomes (Stolze et al., 2015). Recent literature has demonstrated that using pre-trained Large Language Models (LLMs) that encode vast quantities of domain-specific knowledge as the policy offers an exciting approach to exploit the semantic structures in various real-world action spaces (Palo et al., 2023; Zhuang et al., 2024; Hazra et al., 2024). Unfortunately, existing RL frameworks designed to work with LLMs are primarily focused on myopic policies in contextual bandit settings (Ouyang et al., 2024). Recent popular frameworks (von Werra et al., 2020; Hu et al., 2024; Zheng et al., 2024; Harper et al., 2019) mainly focus on techniques for single-turn reinforcement learning. Hence, these existing frameworks can not be directly applied in a nonmyopic BO setting. On the other hand, these methods are unnecessarily complex for various applications in nonmyopic BO. For example, in biological sequence design, a biologist edits specific amino acids in the initial sequence. The transition between consecutive states is determined by these edits, making the process deterministic. Thus, this application does not require a stochastic transition model.

Another limitation of CMBRL is its difficulty in managing reward uncertainties (Ez-zizi et al., 2023). In nonmyopic BO, handling uncertainty is essential (Treven et al., 2024; Sun et al., 2024) for effectively balancing exploitation and exploration (Zangirolami & Borrotti, 2024). Typically, CMBRL algorithms utilize neural reward models which tend to be poorly calibrated (Minderer et al., 2021; Zhao et al., 2024), resulting in overconfident or underconfident reward estimation and potentially leading to suboptimal actions (Sun et al., 2024). To mitigate the limits of exploration in CMBRL and hence the probability of selecting suboptimal actions, recent research emphasizes accounting for reward uncertainties rather than relying solely on average values (Lötjens et al., 2019; Luis et al., 2023; Ez-zizi et al., 2023). This approach enables the use of various acquisition functions to model aleatoric and epistemic uncertainty, allowing policies to better adapt to dynamic or noisy environments, such as biological sequence wet-lab testing, where even minor changes or errors can significantly alter the final results (Caraus et al., 2015).

To address these challenges, we propose a cost-constrained nonmyopic BO algorithm. Our approach reduces the exponential complexity associated with optimizing multiple parameters while maintaining strong exploration capabilities through a computationally efficient, Bayesian reward model. Additionally, this method can be applied across diverse domains, from sequence design to natural language processing, where multiple interactions are required before a final decision is made. Our contributions are summarized as follows.

- We formulate the problem of nonmyopic BO in dynamic cost settings with various cost models inspired by real-world scenarios, such as ones in biological sequence design or geological surveys.

- We propose a novel approach to addressing the nonmyopic BO problem by employing a neural network policy to variationally optimize all decision variables. Our method demonstrates scalability up to a look-ahead horizon of 20 steps, significantly surpassing the state-of-the-art techniques, which typically only extend to four steps within a similar computational budget. Additionally, with an appropriately designed policy network, our method effectively handles a semantically richer action space compared to existing BO methods.

- We extensively benchmark our method against baselines across nine synthetic functions ranging from 2D to 8D, with varying noise levels and requiring 10 to 20 planning steps to find the global optimum. Utilizing a recurrent neural network policy, our approach consistently outperforms traditional acquisition functions. Furthermore, we apply our method to the problem of constrained protein sequence design, where traditional BO is not directly applicable. To facilitate this, we develop an open-source, scalable framework that enhances the efficiency of policy optimization for LLM-based policies within a complex dynamic cost environment. By employing LLaMa-3.2-based policy (Meta, 2024), our method demonstrates superior performance compared to its myopic counterpart, thereby validating its effectiveness.

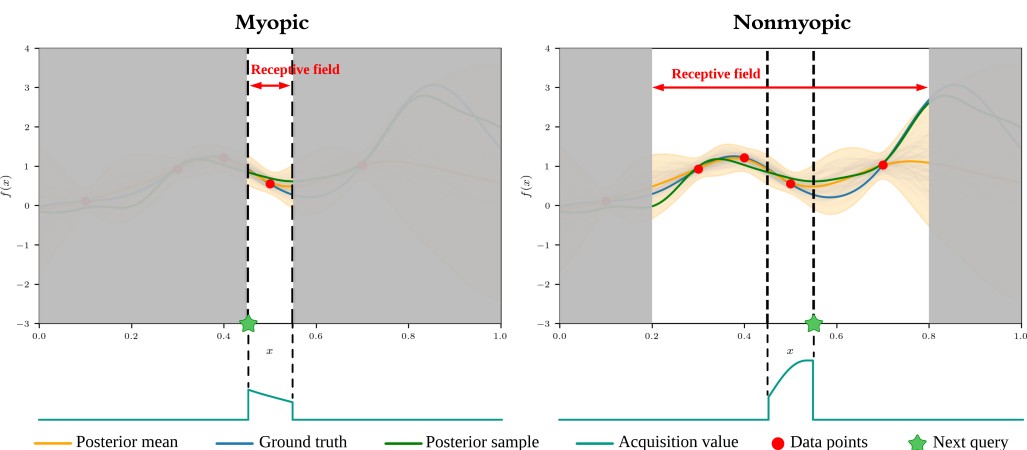

Figure 1: Illustration comparing myopic and nonmyopic decision-making in Bayesian optimization. The top row depicts the receptive field, representing the region of influence for decisions, which varies based on the lookahead horizon. In the myopic setting (left), the receptive field is small, focusing on short-term gain, as indicated by the narrow shaded region. In contrast, the nonmyopic approach (right) accounts for a larger receptive field, favoring long-term gain.

## 2 RELATED WORK

**Nonmyopic Bayesian Optimization in the Dynamic Cost Setting**   Nonmyopic BO has been extensively explored in prior works (Osborne, 2010; González et al., 2016; Wu & Frazier, 2019; Jiang et al., 2020a; Lee et al., 2020; 2021; Astudillo et al., 2021; Folch et al., 2022; Belakaria et al., 2023; Jiang et al., 2020b). These studies focus on converting a nested, multi-step planning problem into a single, high-dimensional optimization problem that can be solved efficiently with quasi-Monte Carlo sampling and gradient-based optimization. The advantage of having a lookahead mechanism is enlarging the receptive field—the area the decision maker can see prior to making a decision (Figure 1). This approach has gained traction in cost-aware and budget-constrained BO (Astudillo et al., 2021; Lee et al., 2021), where nonmyopic planning is crucial. However, a notable challenge with this methodology is its limited scalability when extending the lookahead horizon, primarily due to the exponential increase in the number of decision variables. Our approach introduces a novel combination of Thompson sampling (Thompson, 1933) with the extensive generalization capabilities of a variational network, significantly enhancing the computational efficiency of nonmyopic BO. Utilizing neural networks for variational inference and optimization is not a new concept (Kingma & Welling, 2014; Amos, 2023). For instance, Deep Adaptive Design (Foster et al., 2021; Ivanova et al., 2021) is a parallel line of research from the Bayesian optimal experimental design literature, which has a related but distinct objective to reduce the uncertainty of model parameters as opposed to the global optimization objective in BO. The authors concentrate on reducing the computational demands during deployment and determining the most informative experimental designs by upfront offline optimization of a neural network to amortize the design cost. Our approach diverges by aligning more closely with the principles of online Bayesian optimization. Here, the primary objective extends beyond mere information acquisition to encompass the pursuit of global optimization. Our approach, which employs adaptive decision-making through online policy optimization, could be more robust than offline methods, particularly when the approximation of the reward function changes significantly between queries. This robustness arises because the online policy is updated at each BO step while the offline methods rely on transferring knowledge from learned offline data (Nguyen-Tang & Arora, 2024). The distinctive feature of our study is implementing a pathwise, Thompson sampling-based nonmyopic acquisition function, which significantly reduces the computational cost of the iterative posterior sampling approach in (Jiang et al., 2020b).

**Variational Policy Optimization in Complex Action Spaces**   In many real-world applications, decision-makers must take actions that are complex and subject to semantic constraints. Recent RL research has addressed environments with such actions, which are challenging due to two main

reasons: (i) the large number of potential actions (Hubert et al., 2021; Zhang et al., 2024), and (ii) the complex semantics (Carta et al., 2023) underlying each action, making them difficult to capture. Recent studies have shown that modern LLMs can effectively model semantic actions and be fine-tuned with feedback from the environment (Zhu et al., 2024; Zhang et al., 2024; Zhuang et al., 2024; Hazra et al., 2024). Several papers demonstrate that using LLMs as policy models in reinforcement learning leads to better outcomes (Palo et al., 2023; Zhuang et al., 2024; Hazra et al., 2024). In the field of NLP, a chatbot such as ChatGPT can be viewed as a decision-making process where the underlying LLM must understand user questions or requests to provide appropriate responses. These actions are complex and semantically rich, as even a single word can alter the meaning of a sentence. Consequently, RL methods like proximal policy optimization (Schulman et al., 2017) have been applied to refine the abilities of language models.

**Multi-turn Training Framework for LLMs**   Multi-turn conversations have been shown to be more effective for managing entire dialogues (Zhou et al., 2024). This approach can be viewed as a nonmyopic RL method that trains LLMs to achieve better conversational outcomes. Unfortunately, current RL training frameworks for LLMs, such as TRL (von Werra et al., 2020), OpenRLHF (Hu et al., 2024), LlamaFactory (Zheng et al., 2024), and Nemo (Harper et al., 2019), primarily focus on single-turn conversations. As a result, they are not suited for multi-turn conversation training. When using these frameworks, multi-turn conversations must be divided into individual single turns, which limits the LLM's ability to manage the overall outcome of a conversation effectively.

## 3 METHOD

In this paper, our objective is maximizing an unknown black-box function $f^* : \mathcal{X} \to \mathcal{Y} \subseteq \mathbb{R}$. The decision-maker can make up to $T$ queries $x_{1:T} = [x_1, \ldots, x_T]$ and observe $y_{1:T} = [y_1, \ldots, y_T]$. The output $y_t \in \mathbb{R}$ is observed by evaluating the corresponding query $x_t \in \mathcal{X} = \mathbb{R}^d$ with the homoscedatic noise model $y_t = f^*(x_t) + \epsilon_t$, where $\epsilon_t \sim \mathcal{N}(0, \sigma^2)$. Given a prior distribution over parameters $p(\theta)$, we can sample a probabilistic surrogate model $f_\theta$ of the blackbox function $f^*$ by $\theta \sim p(\theta)$. The posterior distribution of the function conditioned on the history until timestep $t$ is $p_t(\theta) = p(\theta|D_t) = p(\theta|x_1, y_1, ..., x_t, y_t)$. After $T$ queries, the goal is to select an action $a \in \mathcal{A}$ to minimize the $\mathbb{H}_{\ell,c,\mathcal{A}}$-entropy of $f$: $a = \arg\inf_{a \in \mathcal{A}}\{\mathbb{E}_{p_T(f)}[\ell(f, a)] + \lambda c(x_{1:T}, a)\}$. The decision-maker utilizes an acquisition function to make decisions on choosing actions. In this paper, we concentrate on an acquisition function based on decision-theoretic entropy and mutual information, which is known as $H$-Entropy Search (HES) (Neiswanger et al., 2022). Many common acquisition functions, such as Knowledge Gradient (Frazier et al., 2009) and Expected Improvement (Mockus, 1989), can be thought of as a specific case of HES. While HES can tackle various tasks such as top-K or min-max, this study is limited to global optimization of $f^*$ due to resource constraints. Future research will explore these additional tasks. Thus, the action set $\mathcal{A}$ is set to $\mathcal{X}$, and the loss function is defined as $\ell(f^*, a) = -f^*(a)$. Following Russo & Van Roy (2016); Kandasamy et al. (2018), the Bayesian cumulative regret at timestep $T$ is $\mathbb{E}\left[\sum_{t=1}^{T}(f^*(a^*) - f^*(a_t))\right]$, where the integration is over randomness from the environment, queries and final actions. For a prior probability distribution $p(f)$ on functions, along with a dataset $D_t = D_0 \cup \{(x_i, y_i)\}_{i=1}^{t}$ containing observed function evaluations up to $t \in [T]$, we define the entropy of posterior $\mathbb{H}_{\ell,c,\mathcal{A}}$ and the expected $\mathbb{H}_{\ell,c,\mathcal{A}}$-information gain (EHIG) at step $t$ with loss function $\ell$, cost function $c$, action set $\mathcal{A}$, and Lagrange multiplier $\lambda$, and lookahead horizon $L$ as follows (DeGroot, 1962; Neiswanger et al., 2022):

$$\mathbb{H}_{\ell,c,\mathcal{A}}[f|D_t] = \inf_{a \in \mathcal{A}}\{\mathbb{E}_{p_t(f)}[\ell(f, a)] + \lambda c(x_{1:t}, a)\}$$

$$\text{EHIG}_t(x_{1:L}) = \mathbb{H}_{\ell,c,\mathcal{A}}[f|D_t] - \mathbb{E}_{p_t(y_{1:L}|x_{1:L})}[\mathbb{H}_{\ell,c,\mathcal{A}}[f|D_{t+L}]]$$

To minimize the $\mathbb{H}_{\ell,c,\mathcal{A}}$-entropy of $f$, the optimal query $x_{t+1} \in \mathcal{X}$ at each step $t$ must be selected to maximize the expected information gain or reduce uncertainty, as measured by the $\mathbb{H}_{\ell,c,\mathcal{A}}$-entropy. This process involves the decision maker choosing the next query $x_{t+1} = x_1^*$ by optimizing for the

expected reduction in uncertainty as follows.

$$
\begin{aligned}
x_{1:L}^* &= \arg\sup_{x_{1:L}\in\mathcal{X}^L} \text{EHIG}_t(x_{1:L}) = \arg\sup_{x_{1:L}\in\mathcal{X}^L} \left[-\mathbb{E}_{p_t(y_{1:L}|x_{1:L})}[\mathbb{H}_{\ell,c,\mathcal{A}}[f|D_{t+L}]]\right] \\
&= \arg\inf_{x_{1:L}\in\mathcal{X}^L} \left[\mathbb{E}_{p_t(y_{1:L}|x_{1:L})}\left[\inf_{a\in\mathcal{A}}\{\mathbb{E}_{p_{t+L}(f)}[\ell(f,a)] + \lambda c(x_{1:t},x_{1:L},a)\}\right]\right]
\end{aligned}
\tag{1}
$$

### 3.1 BAYESIAN OPTIMIZATION IN DYNAMIC COST SETTINGS

To deal with practical scenarios where the costs of queries change dynamically, we relax the fixed query cost assumption by defining the cost of querying $x_t$ as $c(x_{<t}, x_t)$, where $c$ is an application-specific cost function provided to the decision-maker. The total cost to execute $T$ queries, $x_{1:T}$, is given by $\sum_{t=1}^{T} c(x_{<t}, x_t)$. We define two primary cost structures: (i) Markovian cost, depending only on the previous query, and (ii) non-Markovian cost, depending on the entire query history. The Markovian cost is incurred based on the location of departure $x_{t-1}$ and the destination $x_t$. It also depends on the $p$-norm between $x_t$ and $x_{t-1}$. The relationship between distance and cost in practice can be nonlinear: for example, traveling within a ball of radius of $r$ might be free, but beyond that, the traveling cost grows at a rate of $k$. The observed cost might be perturbed by a random noise $\epsilon$. These ideas are summarized in the following cost model: $c_{\text{Markov}}(x_{t-1}, x_t) = \max(k(\|x_t - x_{t-1}\|_p - r), 0) + \epsilon$. Euclidean cost ($p = 2, r = 0$), Manhattan cost ($p = 1, r = 0$), and $r$-spotlight cost ($k = \infty$) are some commonly used instances. Euclidean cost is found in applications such as ground surveys since the traveling cost depends on the distance between departure and arrival locations (Bordas et al., 2020). Spotlight cost is found in biological sequence design, where editing more than one token is impossible in one experiment (Belanger et al., 2019). In case of non-Markovian cost, the query cost could depend on the entire query history. For example, the traveler in the ground survey application might participate in a mileage point program, where they get a discount $d$ if their total traveling distance is beyond a certain constant $m$. This cost model is represented as $c_{\text{non-Markov}}(x_{<t}, x_t) = c_{\text{Markov}}(x_{t-1}, x_t) - d\mathbb{I}[\sum_{i=1}^{t-1} c_{\text{Markov}}(x_i, x_{i+1}) > m]$. In general, the cost model can be learned from the data provided by the application or designed by the decision-maker. Under a budget constraint commonly found in practice, dynamic costs require efficient nonmyopic planning; otherwise, the next decision may incur a large cost or fail to move beyond local optima.

### 3.2 POLICY NETWORK OPTIMIZATION IN NONMYOPIC BAYESIAN OPTIMIZATION

As the lookahead horizon $L$ increases, the dimension of both optimization and integration increases, making the problem challenging. Below, we apply variational optimization and pathwise sampling to keep the optimization of the above acquisition tractable.

**Variational Optimization** In nonmyopic BO, the number of decision variables is proportional to the number of Monte Carlo samples, which depends on both the number of paths, $p$, and the horizon length, $T$. In the best-case scenario, where the number of Monte Carlo samples grows linearly with the horizon, the policy complexity is $\mathcal{O}(T)$. However, in the worst case, the number of samples grows exponentially, increasing the policy complexity to $\mathcal{O}(k^T)$, where $k$ is the number of samples per step. To address this challenge, we use a variational network to reduce the growth rate of decision parameters with respect to the lookahead horizon from exponential to constant. Variational optimization has been well studied and applied in various contexts, such as policy gradient methods (Schulman et al., 2017), VAEs (Kingma & Welling, 2014), and variational design of experiments (Foster et al., 2019). However, to the best of our knowledge, this approach has not yet been applied in the nonmyopic BO setting to reduce optimization complexity with respect to the lookahead horizon.

From equation 1, we observe that for each lookahead step $l \in [L]$, the optimal decision variable, $x_{t+l+1}^*$, is determined by the previous decision variables and corresponding observations, $(x_{1:t+l}, y_{1:t+l})$. This dependency can be modeled using a recurrent neural network (RNN) parameterized by $\xi \in \Xi$, which takes the history as input to predict the optimal next query: $\xi : (x_{1:t}, y_{1:t}) \mapsto x_{t+1}^*$. The corresponding posterior predictive $y_{t+1}$ can then be computed by $y_{t+1} \sim p_t(x_{t+1}^*)$. Using pathwise sampling, this computation becomes $y_{t+1} = f(x_{t+1}^*)$, where

$f \sim p_t(f)$. Thus, we can maintain gradients for optimizing $x^*_{t+1} = \xi^*(x_{1:t}, y_{1:t})$ across the lookahead steps by applying the chain rule:

$$\frac{\partial \text{EHIG}_t(x_{1:L})}{\partial \xi} = \frac{\partial \text{EHIG}_t(x_{1:L})}{\partial x_{t+L}} \frac{\partial x_{t+L}}{\partial \xi} + \frac{\partial \text{EHIG}_t(x_{1:L})}{\partial y_{t+L}} \frac{\partial y_{t+L}}{\partial x_{t+L}} \frac{\partial x_{t+L}}{\partial \xi}.$$

We can rewrite equation equation 1 as:

$$\xi^* = \arg\inf_{\xi \in \Xi} \left[ \mathbb{E}_{p_t(y_{1:L}|x_{1:L},\xi)} \big[ \inf_{a \in \mathcal{A}} \{ \mathbb{E}_{p_{t+L}(f)}[\ell(f,a)] + \lambda c(x_{1:t}, x_{1:L}, a) \} \big] \right]. \tag{2}$$

In practical applications, $\mathcal{X}$ and $\mathcal{A}$ can be discrete sets. For example, in drug design, molecules can be represented as strings, which can be modeled using NLP techniques by tokenizing them into discrete tokens. The final action in this case might involve deciding whether to accept or reject the designed sequence for wet lab testing. Our method can be extended to handle such discrete cases. One approach is reparameterization by using the reparameterization trick (Kingma & Welling, 2014) or a neural encoder to convert discrete variables into continuous-like ones for simple scenarios, while keeping the rest of the optimization process unchanged. Alternatively, the REINFORCE algorithm (Williams, 1992) can be used to address more complex cases. This algorithm employs the log-derivative trick (Mohamed et al., 2020) to efficiently estimate the optimization gradient as:

$$\frac{\partial \text{EHIG}_t(x_{1:L})}{\partial \xi} = -\frac{1}{L} \sum_{l=1}^{L} \text{EHIG}_t(x_{1:l+1}) \nabla_\xi \log p_t(x_{l+1}|x_{1:l}, \xi).$$

In our experiments, the variational network is trained using imagined data points. Specifically, when optimizing step $t+1$, we use the previously observed data points $(x_{1:t}, y_{1:t})$ to generate imagined lookahead data points $(x_{t+1:t+L}, y_{t+1:t+L})$ through an autoregressive process: $x_{t+l+1} = \xi_t(x_{1:t+l}, y_{1:t+l})$ and $y_{t+l+1} \sim p_{t+l}(x_{1:t+l})$. These imagined data points are then used to compute the optimization objective and find the optimal $\xi_{t+1}$.

**Pathwise Sampling** When the surrogate model is a Gaussian Process (GP), the Monte Carlo method is employed to evaluate the posterior predictive distribution. In prior works, this is done via iterative sampling of the following factorized distribution: $p(y_{1:T}|x_{1:T}, D_0) = \prod_{t=1}^{T} p(y_t|x_t, x_{<t}, y_{<t}, D_0)$. The posterior predictive distribution at the $t$-th step, denoted as $p(y_t|x_t, x_{<t}, y_{<t}, D_0)$, can be approximated by generating $k$ samples of $y_t$ from the GP model. In general, the value of $k$ varies depending on the specific problem. At iteration $t$, suppose that we always sample $k$ samples from the posterior predictive distribution. The number of $y_t$ is $k^t$. This number quickly explodes exponentially with the length of the lookahead horizon (Figure 2). The GP posterior predictive sampling process involves computing the square root of the covariance matrix, which is typically done via Cholesky decomposition. The complexity of this process is proved as $\mathcal{O}(n^3)$ for exact GP or $\mathcal{O}(m^3)$ for approximate GP where $n$ is the total number of samples in the train dataset and $m < n$ is the number of inducing samples (Quiñonero-Candela & Rasmussen, 2005; Wilson et al., 2020). This evidence shows that the complexity for sampling posterior predictive distribution at step $t$-th is at least $\mathcal{O}(k^t m^3)$. One variant of this procedure that can reduce the complexity is limiting the number of sampling samples for posterior predictive approximation at further lookahead steps. For instance, at each step $t > 1$, we can set $k_{t>1} = \max(k_1/2^t, 1)$, where $k_1$ is the predefined number of samples at the first lookahead step. In these cases, we can observe that $\exists \tau : \forall t > \tau, \prod_{t=1}^{T} k_t = K$, where $K$ is a constant. Subsequently, the complexity at step $t$-th can be reduced to $\mathcal{O}(\prod_{t=1}^{T} k_t m^3) = \mathcal{O}(K m^3) = \mathcal{O}(m^3)$.

To mitigate the high complexity of above sampling process, we employ the following factorization: $p(y_{1:T}|x_{1:T}, D_0) = \int p(y_{1:T}|x_{1:T}, f) p(f|D_0) \, df = \int p(f|D_0) \prod_{t=1}^{T} p(y_t|x_t, f) \, df$. The function $f$ is drawn from the prior distribution and path-wise updated via Matheron's rule. For $h$ path, each consists of $T$ steps, the sampling can be done with complexity $\mathcal{O}(h \times T)$. We can approximate the integral arbitrarily well with higher $h$. The gain comes from the fact that we do not need to iteratively compute $K^{-1}_{m,m}$ as in fantasization. If we did, the complexity, with the same number of samples, would be $\mathcal{O}(h \times (T-1)^3)$. This can be done in linear complexity w.r.t. to the number of samples. The complexity of sampling a posterior $\hat{f}$ from $p(f|D_0)$ can be considered as $\mathcal{O}(C)$,

$$y_1^1 \; — \; \cdots \qquad y_2^{2,1} \; — \; \cdots \qquad \cdots \qquad \hat{f}^1 \; — \; x_1 \; — \; y_1^1 \; — \; \cdots$$

$$\boxed{D_0} \; — \; x_1 \; — \; y_1^2 \; — \; x_2^2 \; — \; y_2^{2,2} \; — \; x_3^{2,2} \; — \; \cdots \qquad \boxed{D_0} \; — \; \hat{f}^2 \; — \; x_1 \; — \; y_1^2 \; — \; \cdots$$

$$y_1^3 \; — \; \cdots \qquad y_2^{2,3} \; — \; \cdots \qquad \cdots \qquad \hat{f}^3 \; — \; x_1 \; — \; y_1^3 \; — \; \cdots$$

Figure 2: Posterior predictive sampling (left) and Pathwise sampling (right)

where $C$ is a constant because the number of samples in $D_0$ is unchanged. Then, computing $y_t$ for approximate posterior predictive $p(y_t|x_t, \hat{f})$ can be done by $y_t = \hat{f}(x_t)$, which has complexity of $\mathcal{O}(1)$. Using the same technique as limiting the number of sampling samples, the complexity approximating posterior predictive at any lookahead step is $\mathcal{O}(K)$. Thus, the total complexity at each step $t$-th is $\mathcal{O}(C + K)$. Figure 2 (right) visualizes the concept of this method.

## 4 EXPERIMENTS

In this section, we aim to comprehensively evaluate the performance and robustness of our proposed method. Our experiments are designed to achieve several key goals: first, to demonstrate the algorithm ability to efficiently optimize within environments where query costs vary dynamically; second, to benchmark its performance against established baselines across a range of synthetic and real-world scenarios; and third, to highlight the practical advantages of our approach in terms of both solution quality and computational efficiency. By conducting these experiments, we hope to show that our method can outperform traditional myopic BO techniques and scale effectively to complex, high-dimensional problems.

We compare our method with the following baselines implemented in BoTorch (Balandat et al., 2020). All acquisition function values are estimated via the quasi-Monte Carlo method with the Sobol sequence in BoTorch. We experiment with 4 cost functions: Euclidean, Manhattan, $r$-spotlight, and non-Markovian cost. We use Sample Average Approximation with a base sample as a variance reduction technique that significantly improves the stability of optimization. To enhance the likelihood of convergence, we perform all optimizations using 64 restarts. The lookahead horizon is set to 20 for our method and Multistep Tree. Each experiment is repeated with three random seeds, and all experiments are conducted on an A100 GPU and 80GB of memory.

- Simple Regret (SR) (Zhao et al., 2023) measures the regret or loss in performance between the updated model and the model that would have resulted if the optimal sample had been selected for annotation during the active learning process instead.

- Expected Improvement (EI) (Mockus, 1989) is used to evaluate the usefulness of candidate samples by estimating the expected gain in the performance of a model.

- Probability of Improvement (PI) (Kushner, 1964) calculates the probability of a candidate sample improving the performance of a model compared to the current best sample.

- Upper Confidence Bound (UCB) (Srinivas et al., 2010) balances exploration and exploitation by selecting candidate samples with high uncertainty and high potential for improvement based on the upper confidence bound of their predicted performance.

- Knowledge Gradient (KG) (Frazier et al., 2009) quantifies the expected improvement in the objective function value that would result from evaluating a specific point. It considers the uncertainty of the model predictions and the potential benefit of obtaining additional information about the objective function.

- Multistep Tree (MSL) (Jiang et al., 2020b), which can look up to four steps ahead, is constrained by computational costs. We reimplement this acquisition function using Pathwise sampling, enabling a lookahead horizon of up to 20 steps.

### 4.1 SYNTHETIC FUNCTIONS

We evaluate the performance of our method on nine synthetic functions for global optimization in continuous vector spaces. The 2-dimensional functions, with their initial data points and maximum

BO steps, include Ackley (50 samples, 100 steps), Alpine (100 samples, 50 steps), HolderTable (100 samples, 50 steps), Levy (100 samples, 50 steps), Styblinski-Tang (50 samples, 50 steps), and SynGP (25 samples, 50 steps). The SynGP function is generated from a 2D Gaussian Process with a Radial Basis Function kernel, characterized by a length scale of $\sqrt{0.25}$ and a signal variance of 1. High-dimensional functions include Ackley4D (4D, 100 samples, 100 steps), Hartmann (6D, 500 samples, 100 steps), and Cosine8 (8D, 200 samples, 100 steps). The variation in the number of initial samples and optimization steps reflects the relative complexity of each function. Detailed descriptions of these functions are available in (Bingham, Accessed 2024).

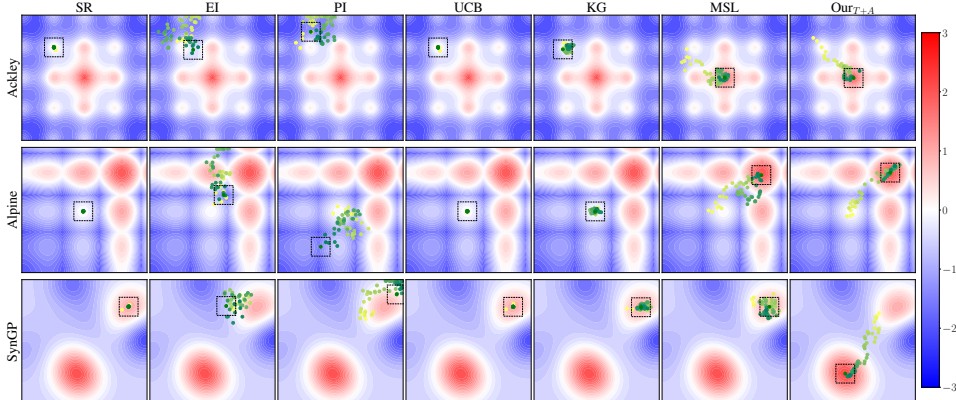

Figure 3: Visualization of queries across BO iterations with setting of $\sigma = 0.0$ and $r$-spotlight cost. The yellow points indicate the starting positions, while the green points represent the final actions. Our method reaches the global optimum, whereas the others tend to be trapped in local optima.

The input of all functions is normalized in a hypercube $[0, 1]^d$, and the output is normalized to the range $-3$ to $3$. Hence, the global maximization of each function is at 3, and the instantaneous regret of action $a$ is $3 - f^*(a)$. The outputs of all functions are observed with three levels of noise: 0%, 1%, and 5%. A Gaussian process with standard Matern kernel is used as the surrogate model. The variational neural network comprises a two-layer encoder, a Gated Recurrent Unit, and a three-layer decoder with exponential linear unit activation functions with 64 hidden dimensions. The network is optimized with Adam optimizer with a learning rate at $10^{-3}$. During inference, we add a small noise sampled from von Mises–Fisher distribution (Fisher, 1953) to the predicted query to enhance the exploration and facilitate acquisition function optimization restart without increasing the number of parameters. Our implementation is available at [omitted for double-blind review].

Figure 3 illustrates that myopic algorithms often converge to local maxima, likely due to their short-sightedness and inability to account for long-term effects, leading to suboptimal solutions that seem beneficial in the short term. In contrast, our proposed method adopts a more foresighted approach, anticipating and considering future outcomes, enabling it to move toward the global maximum. This highlights the inherent advantage of nonmyopic algorithms over myopic ones in pursuing global optimal solutions. The MSL method, with our improvements, can achieve a lookahead of up to 20 steps, yielding outcomes similar to our method. However, without the variational network, MSL optimizes directly on decision variables, which limits its applicability to real-world scenarios, as demonstrated in the next section.

We compare the final observed values between the baseline methods and our proposed at the highest noise level $\sigma = 0.05$ and four cost functions (Figure 4). Our nonmyopic method consistently out-performs the myopic baselines on synthetic functions. The higher values achieved by our method indicate greater predictive accuracy, demonstrating its superior effectiveness. These results sug-gest that the anticipatory capabilities of our method offer a significant advantage, emphasizing the importance of considering future outcomes in algorithmic decision-making. This finding further highlights the need for developing and using nonmyopic algorithms, especially for tasks involving complex, multistep predictions or decisions.

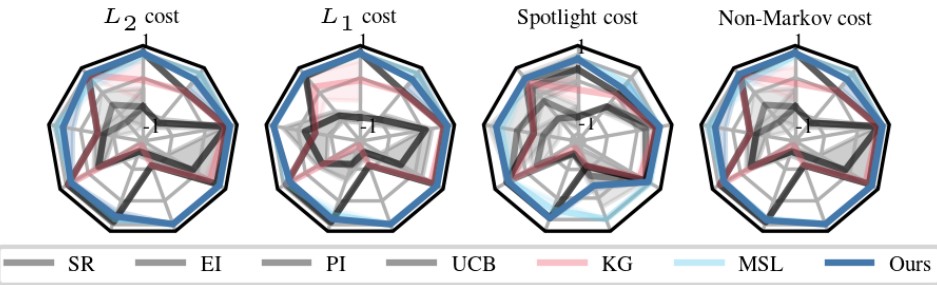

Figure 4: Final observed value at $\sigma = 0.05$. Starting from noon, counter-clockwise: Ackley, Ackley4D, Alpine, Cosine8, Hartmann, HolderTable, Levy, StyblinskiTang, SynGP. We observe that our method consistently achieves the global optimum across various types of cost structures.

## 4.2 Discrete Optimization of Protein Sequence's Fluorescence Level

In this experiment, we apply our method to optimize protein sequences (Elnaggar et al., 2023). Given a protein sequence, the decision maker must decide whether to edit the current sequence. If editing is chosen, the next step is to determine the position to be edited and select the new amino acid. We conduct a sequence of $T = 12$ edits to maximize the fluorescence level obtained from a wet lab experiment, given by the black-box oracle $f^* : \mathcal{X} \to \mathbb{R}$, which is expensive to query. We assume that $f^*$ has a parametric linear functional form on the feature space $\phi(x)$: $y = f_{\theta^*}(x) = g(x) + \alpha(\phi(x)^\top \theta^* + \epsilon)$, where $\theta^* \sim p(\theta) = \mathcal{N}(\mu, \Sigma)$, $\epsilon \sim \mathcal{N}(0, \sigma)$, $g(\cdot)$ is a synthetic function, and $\alpha$ is a scaling hyperparameter. In other words, we select a Gaussian prior for the model parameters and assume a homoscedastic noise observational model.

To build the black-box oracle, we use the ProteinEA Fluorescence dataset (ProteinEA, 2024), which contains 21,445 training samples. We experiment with various featurization functions $\phi(\cdot)$, including Llama2 7B (Touvron et al., 2023), Llama3 8B (Meta, 2024), Mistral 7B (Jiang et al., 2023), Gemma 7B (GemmaTeam, 2024), ESM-2 650M, ESM-2 3B (Lin et al., 2022), and Llama-Molist-Protein 7B (Fang et al., 2024). Figure 15 shows validation results of the parametric black-box oracle with varying training sample sizes. Gemma 7B achieves the highest validation $R^2$ for predicting fluorescence, so we use it as the feature function.

Next, we construct our semi-synthetic protein space using a sequence from the ProteinEA Fluorescence. Specifically, we select a single sequence from the validation set, which consists of 237 amino acids across 20 types. In this experiment, the protein designer can edit only one amino acid at a time across a maximum of 12 fixed positions and is limited to 2 possible amino acid types for each position. Under this setting, the protein space $\mathcal{X}$ contains $|\mathcal{X}| = 4096$ possible proteins. We then compute the fluorescence values for these proteins using the previously constructed oracle. Our goal is to edit a starting protein so that can become the protein with the highest fluorescence, defined as $x_{max} = \arg\max_{x_i \in \mathcal{X}} \mathbb{E}_{\theta^* \sim p(\theta)} f_{\theta^*}(x_i)$. The starting protein, $x_0$, is chosen as the one with an edit distance of 12 from the protein with maximal fluorescence. Because each edit position can only accommodate 2 different tokens in this setup, there is only one possible starting protein. We choose $\alpha = 0.2$ and $g(x) = -0.005(d - 0.5)(d - 5)(d - 8)(d - 13.4)$, where $d = d_{edit}(x, x_0)$ represents the edit distance between the starting protein and a given protein $x$. Figure 5 (left) visualizes the distribution of protein fluorescence values across the sequence of edits.

We employ a Bayesian linear regression model as the surrogate to guide the optimization process, starting with $x_0$. At each step $t \in [T]$, Bayes' rule is used to compute the posterior distribution $p_t(\theta)$, with hyperparameters (e.g., $\sigma$) estimated via maximum marginal log-likelihood. The next candidate mutation is then acquired by optimizing the variational network as follows: $\xi_t = \arg\max_\xi \mathbb{E}_{p_\xi(x_{1:L}|x_0), p_t(\theta)} \left[ f_\theta(x_L) - \lambda \sum_{l=0}^{L-1} c(x_l, x_{l+1}) \right]$. This process seeks to maximize fluorescence while minimizing cost. The next mutation and its fluorescence value are computed by $x_{t+1} \sim p_{\xi_t}(x_t)$ and $f_{\theta^*}(x_{t+1})$, respectively, and the observed dataset is updated accordingly. The optimization then proceeds to the next iteration, and this process can be performed in batches. In this experiment, we use Llama-3.2 3B as the variational network.

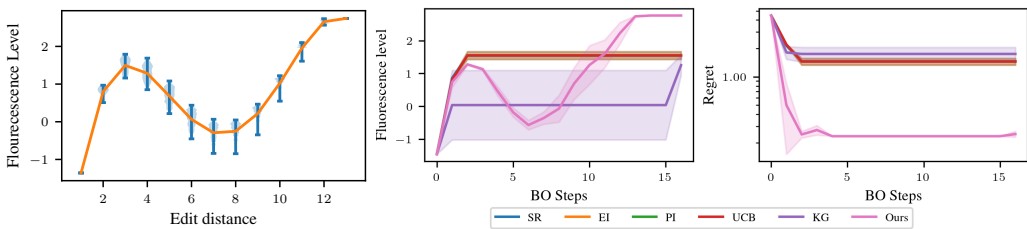

Figure 5: Distribution of fluorescence across edit distance (left). Observed fluorescence levels across BO steps (middle). Regret across BO steps (right).

We set the maximal BO steps to $T = 16$ and the maximum lookahead steps to $L = 12$. Before starting the optimization process, we performed supervised fine-tuning of Llama-3.2 using random mutation data. In each BO iteration, we optimize the network with a maximum of 768 gradient steps and perform 64 restarts to select the next query (i.e., the next mutated protein sequence). To improve the efficiency of the optimization process, we adapted the Proximal Policy Optimization method commonly used in NLP, applying our modifications. Specifically, we separated the lookahead rollout process, using vLLM (Kwon et al., 2023) to infer the variational network more efficiently. After each gradient step, we transfer the network weights to vLLM for the next rollout, ensuring that the upcoming sequences are generated using the latest parameters ($\theta$). To prevent model collapse, the loss function incorporates a KL-divergence term. To handle the constrained cost while generating each lookahead sequence, we attempt to regenerate lookahead sequences 32 times. If it fails, we apply random mutations to the latest sequence, with a 50% chance of no editing. More details about our experiments can be found in Appendix F.

We benchmarked our method against myopic approaches, including SR, EI, PI, UCB, and KG, and conducted experiments using three random seeds. As shown in Figure 5, the myopic methods are trapped and unable to escape local minima, yielding a fluorescence score of approximately 1.5. In contrast, our nonmyopic method, capable of looking ahead 12 steps, was able to anticipate the global maximum region and tolerate lower intermediate values, ultimately reaching a fluorescence level of around 2.7. Figure 5 (middle) shows the observed fluorescence levels, while the (right) side presents the regret ($3 - f_{\theta^*}(a)$) across BO iterations.

## 5 DISCUSSION, LIMITATIONS, FUTURE WORK AND SOCIAL IMPACTS

We have addressed the limitations of traditional myopic BO methods in dynamic cost environments by proposing a nonmyopic approach based on a decision-theoretic generalization of mutual information. Our method incorporates dynamic costs and downstream utility in multistep settings, leading to more informed decision-making under uncertainty. By utilizing a variational network, we achieve scalability in planning multiple steps ahead. Experimental results demonstrate the superior performance of our approach compared to baseline methods, showcasing its potential for various real-world applications. Our work contributes to advancing nonmyopic BO and its practical applicability in dynamic cost scenarios.

While our method offers a significant advancement in nonmyopic Bayesian optimization, it does have certain limitations. Specifically, it requires a well-defined cost model upfront, which can be challenging when costs are uncertain or change dynamically. Furthermore, the performance of our approach is heavily dependent on the accuracy of the underlying Gaussian Process model and the variational network. If these models do not accurately capture the complexities of the underlying black-box function or the cost dynamics, the optimization strategy could underperform.

While our method does not have direct negative social impacts, its misuse or misapplication in specific contexts could raise concerns. For example, if the technique is used in a personalized education setting, it could potentially exacerbate educational inequalities if resources are disproportionately allocated based on the dynamic cost of educating different groups of students. Additionally, there might be ethical implications in healthcare or precision medicine if the dynamic cost leads to bias against certain patient groups due to higher treatment costs. Therefore, it is critical to carefully consider these factors and ensure fair and equitable use of this method across different applications.

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

## A    NOTATION

Table 1 summarizes all the notations used in our paper.

## B    COST STRUCTURE TAXONOMIES

We survey the prior literature on the topic broadly and come up with a taxonomy based on two factors of the cost function: uncertainty and variability. In terms of uncertainty, costs can be classified as known or unknown prior to making a decision. When the cost is unknown, it can be viewed as a random variable that can be modeled probabilistically. In terms of variability, the cost structure can be categorized into dynamic costs, which vary based on the query history, and static costs, which remain fixed for a particular query over time. We note that when the cost is static, the BO community also studies two variations of this structure: heterogeneous cost and homogeneous cost. Homogeneous cost is the setting where the cost of all queries is the same, whereas heterogeneous cost is the setting where the cost of a query is a function of the query itself. Using this taxonomy, we classify the literature into four categories as shown in Table 2.

To further illustrate the distinction between cost structures, we visualize the uncertainty and variability of these structures as probabilistic graphical diagrams (Figure 6). In these diagrams, $f$ represents

Table 1: Notation

| Symbol | Description |
|--------|-------------|
| $f^*$ | Black-box function |
| $p(f)$ | Prior distribution over the black-box function |
| $\theta$ | Random variable representing the parameters of the black-box function in the parametric form |
| $\theta^*$ | Optimal parameters of the black-box function in the parametric form |
| $\mathcal{X}$ | Input domain |
| $\mathcal{Y}$ | Output domain |
| $\phi(\cdot)$ | $\phi : \mathcal{X} \to \mathbb{R}^d$, given $d$-dimensional feature function |
| $\xi$ | Parameters of the variational network |
| $D_t$ | $D_t = \{(x_i, y_i)\}_{i=1}^t$ Dataset acquired |
| $p_t(\cdot)$ | The posterior distribution conditioned on the data up to and including timestep $t$ |
| $c(\cdot, ..., \cdot)$ | $c : \mathcal{X}^k \to \mathbb{R}$, cost function depending on $k$-step history of query |
| $[T]$ | $\{1, ..., T\}$ |
| $\mathbb{H}_{\ell, \mathcal{A}}$ | The decision-theoretic entropy (DeGroot, 1962) corresponding to a loss function $\ell$ and an action set $\mathcal{A}$ |
| $L$ | Lookahead horizon |
| $T$ | Number of interactions with the environment |

Table 2: Comparison of cost types based on uncertainty and variability.

| | **Known cost** | **Unknown cost** |
|---|---|---|
| **Static cost** | Do not vary based on the previous queries and predictable costs, easy to budget over time. 

 Related literature: (Wu & Frazier, 2019; Nyikosa et al., 2018; Lam et al., 2016) | Do not vary based on the previous queries, but the actual amount is not fully known due to external factors. 

 Related literature: (Astudillo et al., 2021; Lee et al., 2021; Snoek et al., 2012; Luong et al., 2021) |
| **Dynamic cost** | Varies based on the previous queries, but can be quantified or predicted. 

 Related literature: Ours | Varies based on the previous queries, and is difficult to predict precisely. 

 Related literature: To the best of our knowledge, we have not yet encountered related work in this category, and we plan to work on this setting in our future direction. |

the target black-box function, $x$ denotes the input query, $y$ is the output value, and $c$ is the cost incurred by querying $x$. On the left — the dynamic-cost structure — the cost of querying $x_3$ can depend on $x_1$ and $x_2$. On the right — the static-known cost structure — the cost of querying $x_3$ is independent of other queries.

Our problem setting focuses on optimizing within a *known and dynamic* cost setting, which is an important cost structure in many practical applications as we motivated earlier. Previous literature has developed methods for complementary-but-distinct cost settings, and we believe that those methods are not suitable for our study as elaborated in the following.

The papers "Multi-step budgeted Bayesian optimization with unknown evaluation costs" (Astudillo et al., 2021), "A nonmyopic approach to cost-constrained Bayesian optimization" (Lee et al., 2021), and "Adaptive cost-aware Bayesian optimization" (Luong et al., 2021) address cost structures characterized by unknown, heterogeneous costs. For instance, in the hyperparameter optimization (HPO) problem studied in these papers, the cost of evaluating a hyperparameter set (i.e., training the target

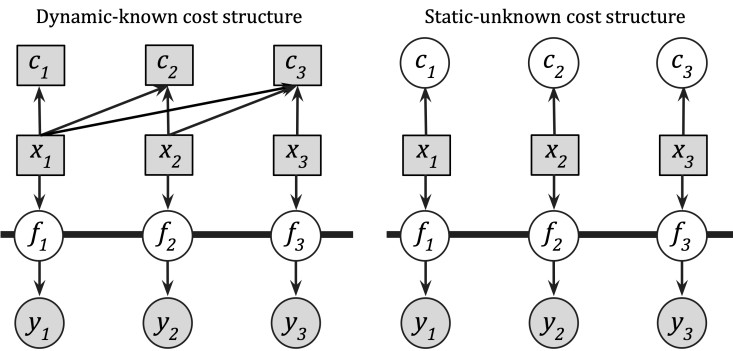

Figure 6: Visualization of known-dynamic and unknow-static cost structures

model with that set) is unknown but static for a particular model and a set of hyperparameters and does not depend on previously chosen sets.

The paper "Bayesian optimization over iterative learners with structured responses: A budget-aware planning approach" (Snoek et al., 2012) also incorporates an unknown, heterogeneous cost structure. The key distinction from the previously mentioned works is the inclusion of an additional cost factor: the number of training epochs for evaluating a hyperparameter set. In this context, for a given location $x$ (a set of hyperparameters) and a specific number of training epochs, the cost is unknown but fixed, as it does not depend on prior queries (previously chosen sets of hyperparameters) or prior choices of training epochs.

Finally, the papers "Practical two-step lookahead Bayesian optimization" (Wu & Frazier, 2019), "Bayesian optimization for dynamic problems" (Nyikosa et al., 2018), and "Bayesian optimization with a finite budget: An approximate dynamic programming approach" (Lam et al., 2016) focus on Bayesian optimization without accounting for cost structures. These works implicitly assume that all locations in the search space have the same, constant, and known cost. This implies that the cost is either zero or any fixed constant and, therefore, not subject to optimization.

Based on the above explaination, the settings in those works are fundamentally different from ours. Regarding the nonmyopic methods presented in the papers (Astudillo et al., 2021; Lee et al., 2021; Snoek et al., 2012; Wu & Frazier, 2019), these approaches extend the Expected Improvement acquisition function to address nonmyopic optimization challenges. While they provide valuable insights, these methods directly optimize free variables in a multi-step tree, which introduces an exponential increase in the number of optimization variables as the lookahead horizon grows. In our context, incorporating their lookahead mechanisms would amount to combining a multi-step tree structure with a dynamic cost function, an approach that is already benchmarked above. The optimization objectives of these lookahead methods, once adapted, align with Equation 1. Our planning algorithm distinguishes itself from the literature by integrating a policy neural network, which is one of our contributions.

## C ABLATION STUDIES

### C.1 NOISE LEVELS

We provide a detailed comparison of our methods and baselines on nine synthetic functions with all cost functions and three noise levels in Figure 7. In this figure, our method demonstrates outstanding performance across all cost structures and noise levels.

### C.2 NUMBER OF INITIAL SAMPLES

We investigated the effect of varying the number of initial samples on the optimization process. Specifically, we evaluated three additional levels of initial samples across three environments: Ackley, Alpine, and SynGP (Figure 8). Our results indicate that with fewer initial points, the GP sur-

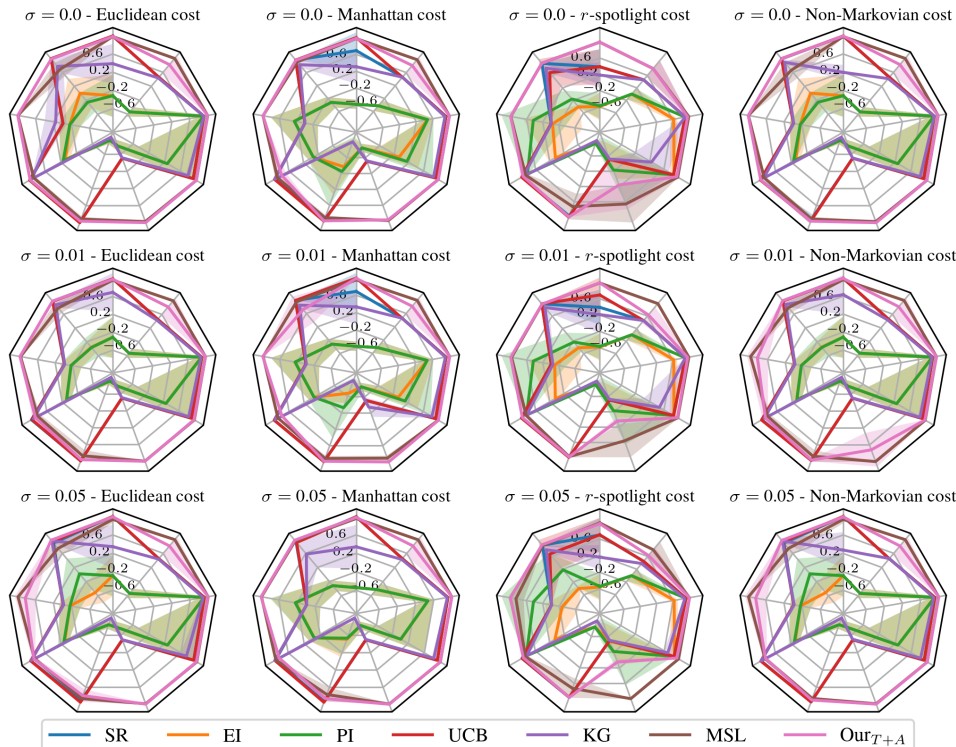

Figure 7: Final observed value. Starting from noon, counter-clockwise: Ackley, Ackley4D, Alpine, Cosine8, Hartmann, HolderTable, Levy, StyblinskiTang, SynGP. We observe that our method consistently achieves the global optimum across various types of cost structures and noise levels

rogate model struggles to accurately approximate the ground-truth function, thereby increasing the likelihood of suboptimal outcomes across both myopic and nonmyopic methods.

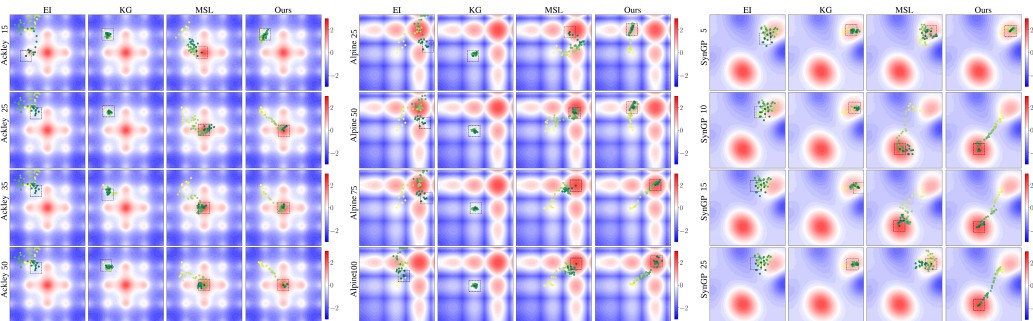

Figure 8: Comparison of performance between our methods and baselines with different number of initial samples. The yellow points indicate the starting positions, while the green points represent the final actions. From top to bottom, the Ackley function is evaluated with 15, 25, 35, and 50 initial samples; the Alpine function with 25, 50, 75, and 100 initial samples; and the SynGP function with 5, 10, and 15 initial samples. With a small number of initial samples, all methods tend to fail to find the global optimum due to poor surrogate models.

## C.3 DIFFERENT KERNELS FOR GP SURROGATE MODEL

To assess the influence of the surrogate model kernel on performance, we evaluated different kernel functions, including the Radial Basis Function (RBF) kernel and the Matérn kernel with $\nu = 1.5$, on three functions: Ackley, Alpine, and SynGP. Figure 9 visualizes the ablation results. This abla-

tion demonstrates that with any well-fitted kernel, the nonmyopic approach can achieve the global optimum.

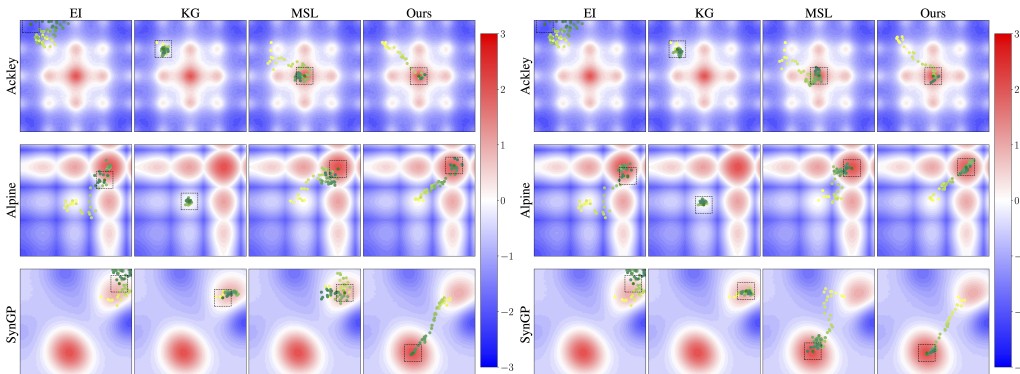

Figure 9: Comparison of performance between our methods and baselines with different kernels for the surrogate model. The yellow points indicate the starting positions, while the green points represent the final actions. The performance of our method is not affected by the choice of kernel for the surrogate model as long as the surrogate model can approximate the target function effectively.

In our synthetic experiments, we do not include an ablation study on Bayesian linear regression model as it is unsuitable for accurately approximating the non-linear target functions. To demonstrate this limitation, we compared the posterior surface generated by Bayesian linear regression with those of other kernel-based methods, as shown in Figure 10. These results confirmed its inadequacy, leading us to exclude it from our ablation study.

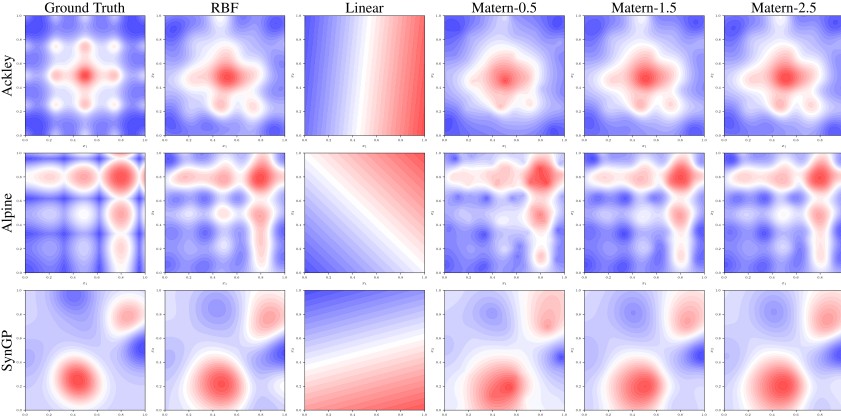

Figure 10: Comparison of posterior surfaces of different kernels on Ackley, Alpine, and SynGP function. Using Bayesian linear regression (the third column) resulted in wrong approximation of the ground truth functions.

## C.4   DIFFERENT LOOKAHEAD STEPS

We included experimental results on the ablation of the number of lookahead steps in Figure 11. These results illustrate the relationship between the number of lookahead steps and the robustness of the optimization, providing insights into how the performance of our approach varies with different horizon lengths. Specifically, with a smaller lookahead horizon, the probability of being trapped by local optima increases, leading to suboptimal optimization in all nonmyopic methods.

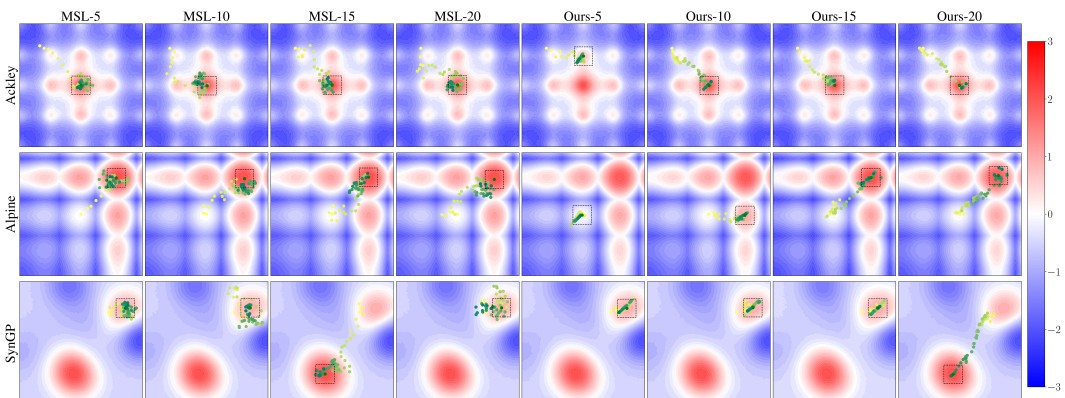

Figure 11: Comparison of our method and nonmyopic baseline at 5, 10, 15, and 20 lookahead steps. The yellow points indicate the starting positions, while the green points represent the final actions. With fewer lookahead steps, nonmyopic methods tend to fail to find the global optimum, demonstrating the benefit of having a longer lookahead horizon.

## D  ABLATION STUDY ON MYOPIC METHOD

Regarding using the UCB acquisition function, the level of optimism can be controlled by the $\beta$ hyperparameter. A smaller $\beta$ prioritizes exploitation, while a larger $\beta$ prioritizes exploration. With sufficiently large $\beta$, the standard deviation term dominates the mean term, leading to decisions driven by the most uncertain areas. To further illustrate the impact of large $\beta$, we conducted additional experiments with $\beta$ values ranging from 0.1 to 1000 on nine synthetic functions. In Figure 12 we highlight the behavior of UCB when increasing $\beta$.

We also provide the value of the final action, normalized to range from -1 to 1, where -1 represents the worst outcome and 1 is the best in Table 3, 4, and 5. These empirical results further illustrate that the large $\beta$ value can encourage the decision maker to make queries that highly prioritize exploration. As illustrated in the above figure and table, such exploration are typically myopic and unplanned, and consequently, the decision maker typically misses the global optima or overexplore the un-promising region. We also want to note that in our experiment, no single $\beta$ outperformed others in all settings: for example, $\beta = 10$ works well for Ackley, but does not work for other functions. Indeed, choosing the value of $\beta$ for UCB before running the online experiment is nontrivial in practice.

Table 3: Comparison of final action value of our method with lookahead 20 steps and UCB with various $\beta$ (part 1)

|  | Ours | UCB ($\beta = 0.1$) | UCB ($\beta = 0.5$) | UCB ($\beta = 1$) | UCB ($\beta = 2$) |
|---|---|---|---|---|---|
| Ackley | 0.97±0.03 | 0.7±0.4 | 0.4±0.41 | 0.4±0.41 | 0.4±0.41 |
| Ackley4D | 0.97±0.02 | 0.68±0.44 | 0.68±0.44 | 0.67±0.43 | 0.68±0.44 |
| Alpine | 0.99±0.0 | -0.01±0.01 | -0.01±0.01 | -0.01±0.01 | -0.01±0.01 |
| Cosine8 | 0.93±0.01 | 0.96±0.01 | 0.97±0.01 | 0.96±0.02 | 0.97±0.02 |
| Hartmann | 0.96±0.03 | 0.95±0.02 | 0.94±0.04 | 0.95±0.03 | 0.95±0.03 |
| HolderTable | 0.05±0.08 | -0.49±0.01 | -0.5±0.01 | -0.49±0.01 | -0.49±0.01 |
| Levy | 0.95±0.0 | 0.87±0.0 | 0.87±0.0 | 0.87±0.0 | 0.87±0.0 |
| StyblinskiTang | 1.0±0.0 | 0.91±0.0 | 0.91±0.0 | 0.91±0.0 | 0.91±0.0 |
| SynGP | 0.63±0.25 | 0.45±0.01 | 0.45±0.01 | 0.45±0.01 | 0.45±0.01 |

## E  REAL EXPERIMENTS ON CONTINUOUS SPACE

To demonstrate the applicability of our methods in real-life continuous environments, we conducted additional experiments on human travel optimization in a 2D continuous domain. Specifically, we

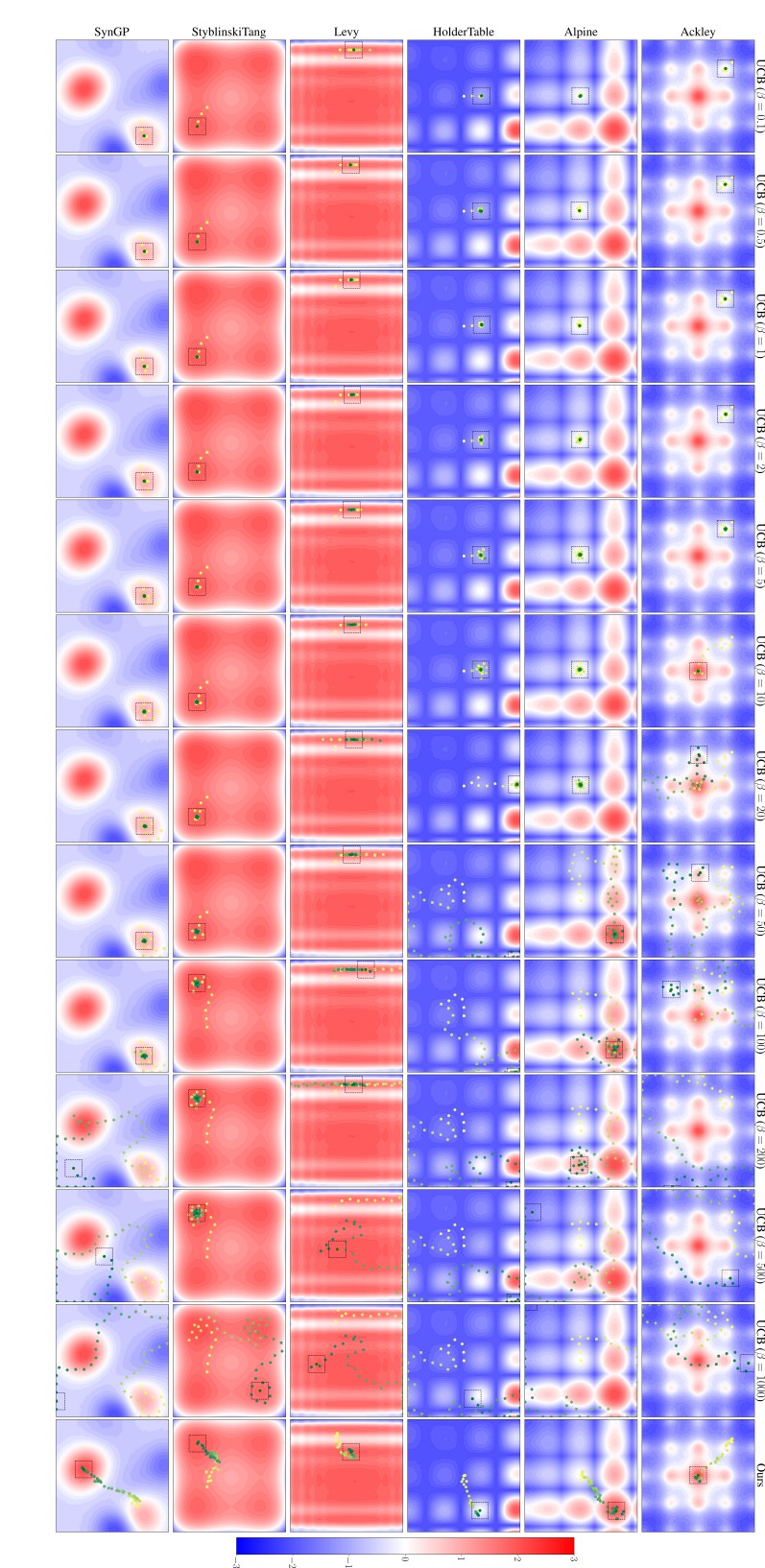

Figure 12: Visualization of queries across BO iterations with the setting of $\sigma = 0.0$ and r-spotlight cost. The yellow points indicate the starting positions, while the green points represent the final actions. With appropriate $\beta$, the UCB can achieve the global optimum as ours.

Table 4: Comparison of final action value of our method with lookahead 20 steps and UCB with various $\beta$ (part 2)

|  | UCB ($\beta = 5$) | UCB ($\beta = 10$) | UCB ($\beta = 20$) | UCB ($\beta = 50$) |
|---|---|---|---|---|
| Ackley | 0.7±0.43 | 1.0±0.01 | 0.71±0.35 | 0.55±0.65 |
| Ackley4D | 0.98±0.01 | 0.98±0.01 | 0.87±0.17 | 0.99±0.01 |
| Alpine | -0.01±0.01 | -0.01±0.01 | 0.15±0.24 | 0.73±0.36 |
| Cosine8 | 0.96±0.03 | 0.97±0.02 | 0.97±0.04 | 0.88±0.04 |
| Hartmann | 0.97±0.02 | 0.96±0.03 | 0.97±0.03 | 0.94±0.03 |
| HolderTable | -0.49±0.01 | -0.28±0.32 | -0.03±0.34 | 0.81±0.6 |
| Levy | 0.87±0.0 | 0.87±0.0 | 0.87±0.0 | 0.87±0.0 |
| StyblinskiTang | 0.91±0.0 | 0.91±0.0 | 0.91±0.0 | 0.91±0.0 |
| SynGP | 0.45±0.01 | 0.45±0.01 | 0.45±0.01 | 0.45±0.01 |

Table 5: Comparison of final action value of our method with lookahead 20 steps and UCB with various $\beta$ (part 3)

|  | UCB ($\beta = 100$) | UCB ($\beta = 200$) | UCB ($\beta = 500$) | UCB ($\beta = 1000$) |
|---|---|---|---|---|
| Ackley | 0.45±0.41 | -0.34±0.38 | -0.12±0.13 | 0.06±0.5 |
| Ackley4D | 0.47±0.33 | -0.72±0.06 | -0.33±0.3 | -0.6±0.24 |
| Alpine | 0.92±0.09 | 0.15±0.46 | -0.43±0.48 | -0.2±0.66 |
| Cosine8 | 0.71±0.06 | 0.62±0.07 | 0.13±0.18 | -0.18±0.1 |
| Hartmann | 0.95±0.03 | 0.81±0.12 | -0.02±0.61 | -0.52±0.43 |
| HolderTable | 0.9±0.81 | 0.76±0.23 | 0.27±1.1 | -0.41±0.32 |
| Levy | 0.86±0.01 | 0.93±0.05 | 0.82±0.12 | 0.84±0.06 |
| StyblinskiTang | 0.94±0.04 | 0.93±0.1 | 0.96±0.06 | 0.92±0.07 |
| SynGP | 0.45±0.01 | 0.22±0.32 | 0.61±0.53 | 0.5±0.53 |

utilized an image from NASA's Earth Observatory[1], which is a 2016 grayscale image of night lights in Georgia and South Carolina states, with a resolution of $1000 \times 1000$ pixels. To facilitate the optimization of the GP surrogate model and avoid numerical issues due to image noise, we applied a stack blur with a radius of 40 to the image. The pixel values, ranging from 0 to 255, were normalized to a range of $-3$ to 3. The image width and height were normalized to a range of 0 to 1. We apply our methods and baselines with spotlight cost ($r = 0.1$) and Euclidean cost. Figures 13 and 14 show the results of our methods and baselines on spotlight and Euclidean cost, respectively. In this environment, nonmyopic methods demonstrated their advantage in lookahead capability. Notably, our method showed its effectiveness in directly reaching the global optimum, rather than querying around sub-optimal locations before approaching the global optimum as MSL.

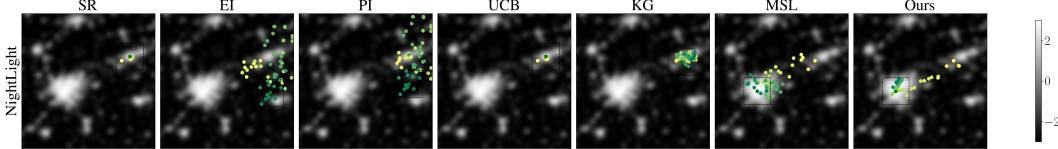

Figure 13: Visualization of different methods on NASA night light images in the case of spotlight cost.

# F  DETAILS OF PROTEIN SEQUENCE DESIGN EXPERIMENT

## F.1  ORACLE GOODNESS OF FIT

We present the goodness of fit for various featurization functions in Figure 15. The $R^2$ metric is used to evaluate the performance of embedding protein sequences.

---

[1]https://earthobservatory.nasa.gov/features/NightLights

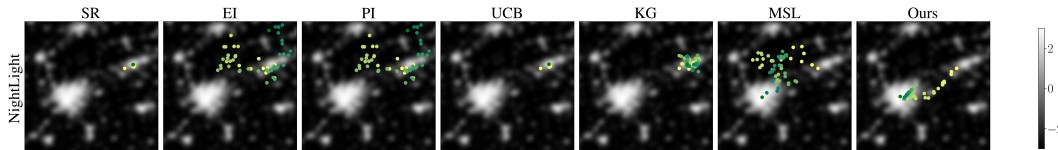

Figure 14: Visualization of different methods on NASA night light images in the case of Euclidean cost.

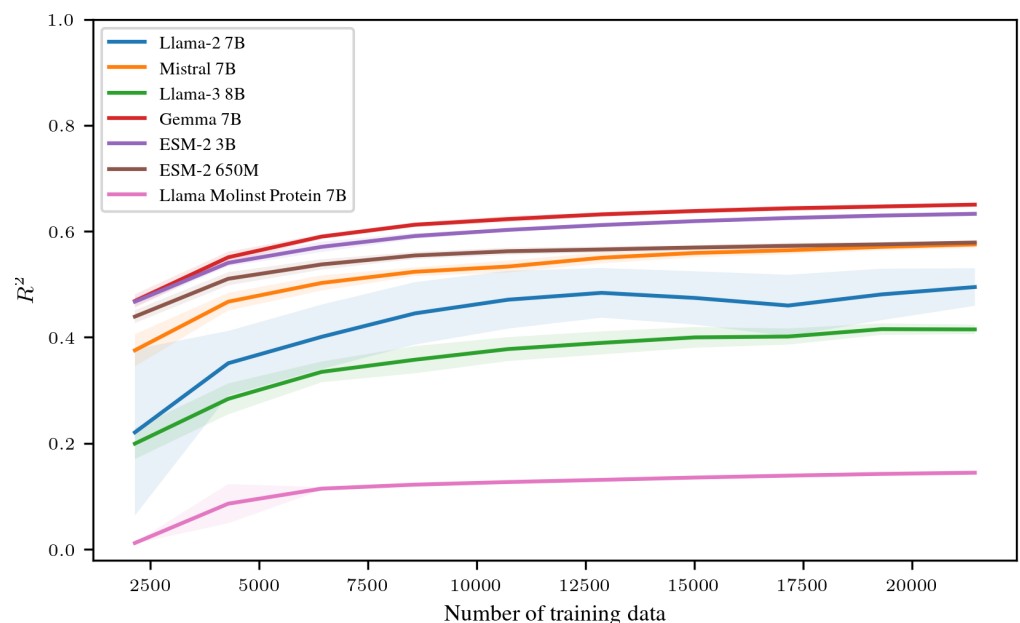

Figure 15: Goodness of fit on the test set as a function of training data

### F.2 MODELING PROTEIN DESIGN AS A CHAT CONVERSATION

We employ the instruction-finetuned Llama-3.2 3B[2] model as our variational network. To leverage the model's conversational capabilities, we frame the protein design process as a dialogue. Specifically, we prompt the model to generate the next protein sequences. The prompts we used are outlined below.

System prompt:

```
You are a helpful assistant who works in a protein engineering lab.
We are trying to edit a given protein by a sequence of 1-step protein
editing, known as mutation. You need to use your knowledge to help me
propose suitable protein editing. Going from an initial protein to an
optimal one can take many steps.
```

First prompt:

```
Edit 1 amino acid in the below protein sequence to create a new
protein with higher fluorescence. The amino acid must be in set {D, E}.
Protein sequence: {starting_protein}
```

Feedback prompt:

```
Fluorescence level of the above protein: {fluorescence_level}
```

---

[2]https://huggingface.co/meta-llama/Llama-3.2-3B-Instruct

```
Based on the above protein sequence and its fluorescence value, edit
1 amino acid to achieve higher fluorescence. You must only return the
modified protein sequence and nothing else.
Modified protein sequence:
```

### F.3 SUPERVISED FINE-TUNING PROCESS

Before initiating the BO process, we perform supervised fine-tuning (SFT) on the variational network to familiarize it with the protein design task. We generate a dataset for SFT training consisting of 100 dialogues, each containing $L$ rounds corresponding to the number of lookahead steps. The proteins in each dialogue are created by either randomly mutating or retaining the previous protein. The fine-tuning hyperparameters are provided below.

- Learning rate: $10^{-4}$
- Epochs: 3.3
- Batch size: 4
- Learning rate warmup ratio: 0.1
- Learning rate scheduling: Cosine
- LoRA $\alpha$: 32
- LoRA $r$: 16
- LoRA dropout: 0.1
- LoRA target modules: q_proj, v_proj

### F.4 NONMYOPIC BO AS MULTI-TURN PPO FINETUNING

Proximal Policy Optimization (PPO) (Schulman et al., 2017) is typically used to fine-tune language models for single-turn conversations, where the model responds once to a prompt without considering future turns. However, our approach requires the model to think ahead and generate multiple future queries (in this case, protein sequences) over several turns. To address this, we modify existing PPO frameworks to handle multiturn conversations, allowing the model to generate and optimize future sequences during training. We also use vLLM (Kwon et al., 2023), a system designed to improve the speed and efficiency of inference (i.e., generating outputs from the model). However, vLLM is built for inference only and cannot be used directly for training. To overcome this, after each step of updating the model during training (called a gradient step), we transfer the updated model's weights (parameters) to the vLLM system. This allows us to use vLLM for faster generation of outputs, leading to more efficient training.

In the PPO training process, we calculate a final reward for each dialogue using a function $\ell$. This function varies depending on the acquisition method being used (e.g., expected improvement or simple regret). Once the reward is computed, it is adjusted, or "discounted," for each individual turn in the dialogue. This means that actions taken earlier in the conversation get less reward compared to later actions. We then use this discounted reward as feedback to update the model during PPO training. By doing this, we extend the single-turn PPO framework, which normally handles one response at a time, to work for our multiturn conversation data. The hyperparameters used for fine-tuning PPO are provided below.

- Learning rate: $10^{-4}$
- Epochs: 64
- Batch size: 1
- Learning rate warmup ratio: 0.1
- Learning rate scheduling: Cosine
- LoRA $\alpha$: 256
- LoRA $r$: 128
- LoRA dropout: 0.1

- LoRA target modules: q_proj, v_proj
- Maximal rollout retry: 32
- Discount reward factor: 0.95

# G  ABLATION STUDY ON PROTEIN DESIGN EXPERIMENTS

## G.1  VARYING PROTEIN SPACE

We ablate three different starting proteins and two different synthetic functions $g(x)$ to construct protein spaces. Visualizations of these protein spaces are presented in Figure 16.

Table 6: Protein space constraints

| No. | Starting protein | Allowed positions | Allowed AAs |
|---|---|---|---|
| 1 | SKGEELFTGVVPILVELGGDVNGHKFSVSGEGEG DATYGKLTLKFICTTGKLPVPWPTLVTTLSYGVQ CFSRFPDHMKQHDFFKSAMPEGYVQERTIFSKDD GNYKTRAEVKFEGDELVNRIELKGIDFKEEENILG HKLEENYNSHNVYIMADDQKNGIKVNFKIRHNIE DDSVQLADHYQQNTPIGDEPVLLPDDHYLSTQSA LSKDDNEDRDEMVLLEFVTAAGITHGMDELYK | 116, 131, 132, 141, 154, 171, 172, 189, 196, 209, 212, 215 | E, D |
| 2 | SKPEELFTPVVGILVELDPDVNGHKFSVSGEGEPD ATYGKLTLKFICTTGKLGVGWGTLVTTLSYGVQC FSRYPDHMKQHDFFKSAMPEGYVQERTIFFKDDG NYKTRAEVKFEPDTLVNRIELKGIVFKEDGNTLG HKLEYNYNSHNVYIMADEQKNGIKVNFKIRHNIE DGSVQLADHYQQNTPIPDGPVLLPDNHYLSTQSA LSKDPNEKRDHMVLLEFVTAAGITHGMDELYK | 2, 8, 11, 18, 33, 52, 54, 56, 114, 158, 187, 190 | G, P |

$$g_1(x) = -0.005(d - 0.5)(d - 5)(d - 8)(d - 13.4)$$

$$g_2(x) = -e^{-0.7 \cdot \sqrt{0.5 \cdot d^2}} - e^{0.5 \cdot \cos(0.4\pi d)} + e + 0.3$$

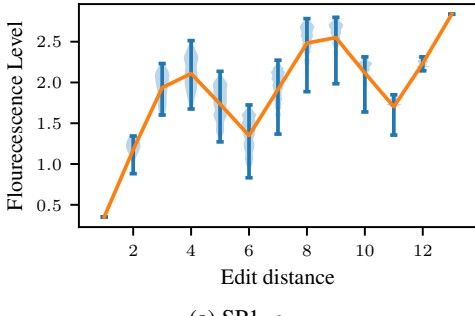

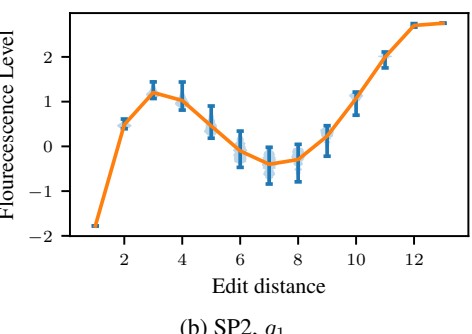

(a) SP1, $g_2$          (b) SP2, $g_1$

Figure 16: Ablation of protein spaces with a different starting protein (SP) and a different synthetic function

## G.2  RESULTS OF DIFFERENT PROTEIN SPACES

We present the results of additional experiments on protein design with the same starting protein with $g_2$ (Figure 17), and with a different starting protein with $g_1$ (Figure 18). These figures demonstrate that our proposed nonmyopic method outperforms other myopic baselines in

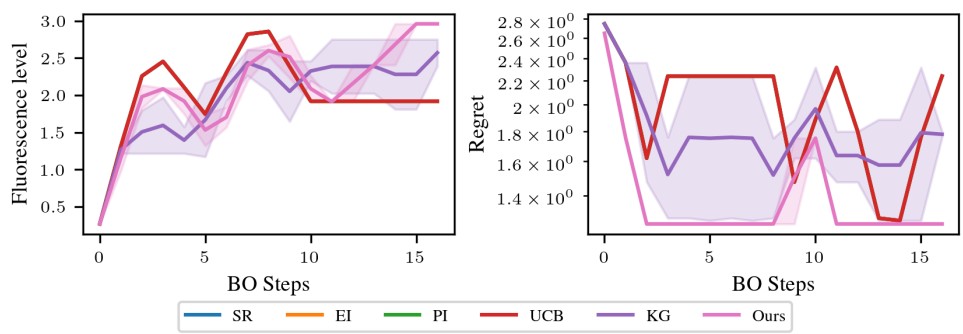

Figure 17: Caption

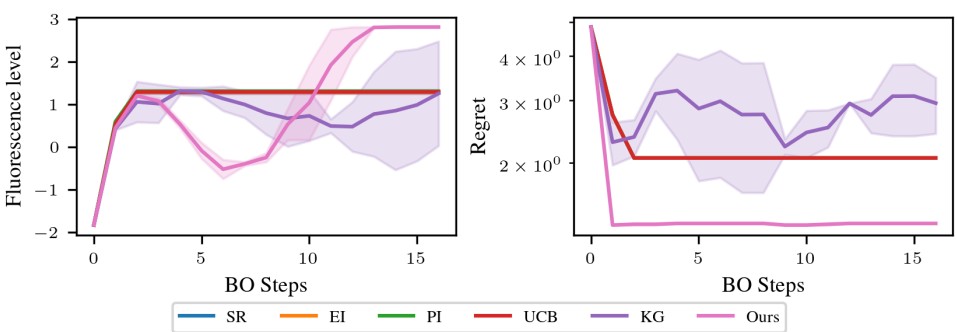

Figure 18: Caption

## H   VISUALIZATION OF PROTEIN EDITING RESULT

We visualized the designed proteins with the first starting protein and $g_1$. We use ESMFold (Lin et al., 2022) to fold the designed proteins and PyMol (Schrödinger, LLC, 2015) to visualize them. The visualizations are presented in Table 7.

Table 7: Visualization of designed proteins

| Protein | Visualization |
|---|---|
| **Starting protein:**

SKGEELFTGVVPILVELGGDVNGHKFSVSGEGEGDATY GKLTLKFICTTGKLPVPWPTLVTTLSYGVQCFSRFPDHM KQHDFFKSAMPEGYVQERTIFSKDDGNYKTRAEVKFEG DELVNRIELKGIDFKEEENILGHKLEENYNSHNVYIMAD DQKNGIKVNFKIRHNIEDDSVQLADHYQQNTPIGDEPVL LPDDHYLSTQSALSKDDNEDRDEMVLLEFVTAAGITHG MDELYK |  |
| **Our - Optimal:**
SKGEELFTGVVPILVELGGDVNGHKFSVSGEGEGDATY GKLTLKFICTTGKLPVPWPTLVTTLSYGVQCFSRFPDHM KQHDFFKSAMPEGYVQERTIFSKDDGNYKTRAEVKFEG DDLVNRIELKGIDFKEDDNILGHKLEDNYNSHNVYIMA DEQKNGIKVNFKIRHNIEEESVQLADHYQQNTPIGDDPV LLPDEHYLSTQSALSKDENEERDDMVLLEFVTAAGITHG MDELYK |  |
| **SR:**

SKGEELFTGVVPILVELGGDVNGHKFSVSGEGEGDATY GKLTLKFICTTGKLPVPWPTLVTTLSYGVQCFSRFPDHM KQHDFFKSAMPEGYVQERTIFSKDDGNYKTRAEVKFEG DELVNRIELKGIDFKEEDNILGHKLEENYNSHNVYIMAD DQKNGIKVNFKIRHNIEDDSVQLADHYQQNTPIGDDPVL LPDDHYLSTQSALSKDDNEDRDEMVLLEFVTAAGITHG MDELYK |  |
| **EI, PI, UCB:**
SKGEELFTGVVPILVELGGDVNGHKFSVSGEGEGDATY GKLTLKFICTTGKLPVPWPTLVTTLSYGVQCFSRFPDHM KQHDFFKSAMPEGYVQERTIFSKDDGNYKTRAEVKFEG DELVNRIELKGIDFKEDENILGHKLEENYNSHNVYIMAD DQKNGIKVNFKIRHNIEDDSVQLADHYQQNTPIGDEPVL LPDDHYLSTQSALSKDENEDRDEMVLLEFVTAAGITHG MDELYK |  |
| **KG:**

SKGEELFTGVVPILVELGGDVNGHKFSVSGEGEGDATY GKLTLKFICTTGKLPVPWPTLVTTLSYGVQCFSRFPDHM KQHDFFKSAMPEGYVQERTIFSKDDGNYKTRAEVKFEG DELVNRIELKGIDFKEEENILGHKLEENYNSHNVYIMAD DQKNGIKVNFKIRHNIEDDSVQLADHYQQNTPIGDEPVL LPDDHYLSTQSALSKDDNEERDEMVLLEFVTAAGITHG MDELYK |  |

