# OpenReview forum: "Nonmyopic Bayesian Optimization in Dynamic Cost Settings"
_ICLR.cc/2025/Conference — Submitted to ICLR 2025_

### Official Review · Reviewer_aZur · 2024-10-31

**Soundness:** 2
**Presentation:** 2
**Contribution:** 2
**Rating:** 3
**Confidence:** 3

**Summary:**

The work studies the problem of performing (global) optimization of a black-box function, where the cost between actions may vary. The work proposes to learn a neural network policy via variational optimization of a multi-step objective. The authors conduct experiments on a large number of synthetic test functions, as well as in a protein design task whose setting appears slightly different from the main work.

**Strengths:**

The paper aims to address the real problem of dynamic cost settings during black-box optimization.
The proposed algorithm appears to work across a range of tasks.

Moreover, the paper may provide preliminary supporting evidence that multi-step planning can improve performance over single-step algorithms.

The limitations and potential ethical impacts are well-discussed.

**Weaknesses:**

* The paper in its current form does not make a comprehensive argument why multi-step planning may be preferable over single-step planning with optimism. Most single-step algorithms for BO such as the evaluated EI and UCB crucially rely on optimism to explore and prevent getting stuck at a local optimum. The evaluation of these baselines is limited (as I discuss in the following) and in its current form, is not fully convincing.

* While one of the key motivations behind this work, appears to be the practical importance of dynamic costs, the paper does not really address this in its evaluation.
While the proposed algorithm, balances performance and cost via the hyperparameter $\lambda$, while the employed performance measure (Bayesian cumulative regret) does not take into account cost at all. It is not clear how costs are evaluated within the experiments of Section 4.1. Furthermore, based on this, it also unclear how baselines take into account costs. The authors mention that the evaluate across multiple hand-designed costs, but how these costs are reflected in the experiment is not discussed.

Experiments on synthetic functions:

* In Figure 2, the various approaches are evaluated with *different* initial conditions. In my opinion, the initial conditions would have to be fixed across all methods for the results to be interpretable.

* A more general concern I have with the experiments is the evaluation of the baselines, in particular, the hyperparameter choices. It is well known that BO methods are sensetive to their hyperparameters to ensure sufficient levels of "optimism" not to get stuck in local optima (e.g., the $\beta$ in UCB). Moreover, the choice of kernel (*output scale & length scale*) critically influences whether methods will get "stuck" or find the global optimum. I did not find any details regarding this in the paper, nor any ablation studies. In my opinion, those are needed to be able to trust the results.

Experiments on protein design: I find the discussion of the experimental setup not very detailed (see my questions) and not related enough to preceding sections to interpret the results. Overall, the setting appears to depart significantly from the previously discussed setting and several details of the approach remain unclear.



More minor pointers:

* The key novelty the paper claims about the proposed algorithm is the application of variational optimization to non-myopic BO. The paper cites [1] as reference for variational optimization of designs (line 241), but should cite [2]. Moreover, [1] which is a follow-up to [2], proposes a non-myopic approach to experiment design. This present paper claims to be "more robust" than their approach (DAD), as far as I can see, without presenting empirical evidence.

* The paper discusses that computing the GP posterior has cubic complexity in the number of inducing points, but this can be reduced when using Bayesian linear regression with suitable features (e.g., random Fourier features, [3]) which can approximate the GP posterior to an arbitrary accuracy.

[1]: Foster et al. Deep adaptive design: Amortizing sequential bayesian experimental design.
[2]: Foster et al. Variational bayesian optimal experimental design.
[3]: Rahimi et al. Random Features for Large-Scale Kernel Machines.

**Questions:**

* The paper discusses the idea of "pathwise sampling" in lines 276-. Can the authors clarify how this approach of sampling a function from the probabilistic model and then optimizing a sequence of steps over this (now fixed) function is different from trajectory sampling [1] used in PETS, with similar ideas being used in model-based RL with PILCO [2]?

* Why did the authors choose 64 restarts? Are the results meaningfully different with a different number of restarts? Requiring such a large number of restarts seems like a drawback of the proposed approach. Moreover, it would benefit the manuscript to explain what is meant by a "restart"; I assume it refers to starting the optimization process from multiple initial conditions.

* It is mentioned that the MSL method is not applicable to "real-world" tasks and referred to Section 4.2 (line 395), but I do not find any reference to MSL in the following.

Questions regarding experiments on protein design:

* Given that predictions of an LLM are used as the "ground-truth" fluorescence levels, could the protein sequence with large edit distance (Figure 4) be outliers / wrong predictions of the LLM? Otherwise, what are alternative explanations for the S-shaped curve in Figure 4?
* How are the hyperparameters chosen? Are they chosen on the validation set over which is also optimized?
* Why is the edit distance represented in the reward value (via $g$) and not only in the cost?
* How is $\lambda$ chosen?
* How does the regret reflect the cost of the chosen protein? That is, in what sense does the cost function $c$ affect the outcome of the empirical evaluation at all?
* It is not clear to me how the authors optimize the variational objective with the *non-Bayesian* LLM in this setting. Can the authors clarify in more detail how this is related to the algorithm proposed in the main text?
* It is mentioned that steps sometimes "fail" (line 494). Can the authors elaborate what is meant by this and clarify how often it fails for each of the discussed baselines?
* The paper provides no details on how the baselines are evaluated in this setting. Can the authors clarify this? In particular, can the authors clarify how "uncertainty" is measured in this context when we do not have a Bayesian model (as in the case of Llama-3.2). Or is a Bayesian linear regression uses, in which case I wonder how this can be compared to the Llama-3.2 policy network used in the proposed approach?

[1]: Chua et al. Deep Reinforcement Learning in a Handful of Trials using Probabilistic Dynamics Models.
[2]: Deisenroth et al. PILCO: A Model-Based and Data-Eﬃcient Approach to Policy Search.

---

> ### Author Response · Authors · 2024-11-23
> **Response for weaknesses**
>
> Dear Reviewer aZur,
>
> Thank you for raising the need for *a comprehensive argument regarding the advantages of multi-step planning over single-step planning with optimism*. We appreciate the opportunity to provide a more detailed explanation and address your concerns. To illustrate our position, consider the example of optimizing a 1D function using a myopic approach (UCB) and our non-myopic method with a 6-step lookahead, as shown in the figure below. The top row highlights the receptive field - the region of influence for decisions—which varies with the planning horizon. In the myopic setting (left), the receptive field is narrow, focusing on short-term gains, as indicated by the limited shaded region. In contrast, the non-myopic approach (right) extends the receptive field, enabling the method to account for longer-term outcomes.
>
> [Figure](https://anonymous.4open.science/r/nonmyopia/images/ucb/NonmyopicBO%20-%20UCB.png)
>
> Both methods operate within the same radius (depicted as the area between the two dashed lines in the top row), but the key difference lies in the non-myopic method’s broader receptive field. This expanded perspective allows the method to better anticipate and account for future rewards. In contrast, the myopic approach, while potentially equipped with prior knowledge of the global optimum, prioritizes immediate rewards. This short-term focus may prevent it from overcoming local valleys to reach the global optimum.
>
> Secondly, in our experiments on synthetic functions, we conducted additional ablation studies to rigorously evaluate the effectiveness of non-myopic planning under varying initial conditions and kernel choices. Additionally, we analyzed the impact of hyperparameter configurations for the UCB to demonstrate its susceptibility to local optima.
>
> 1. *Number of Initial Samples:* We investigated the effect of varying the number of initial samples on the optimization process. Our results show that with fewer initial points, the surrogate model struggles to approximate the ground-truth function accurately, increasing the likelihood of suboptimal outcomes across all methods.
> [Figure for different inital samples](https://anonymous.4open.science/r/nonmyopia/images/initial_samples/NonmyopicBO%20-%20Initial_samples.png)
>
> 2. *Kernel Choices:* To assess the influence of the surrogate model’s kernel on performance, we evaluated different kernel functions, including the RBF kernel and the Matern kernel with $\nu = 1.5$. With this ablation, we can observe that with any well-fitted kernels, the nonmyopic approach can achieve the global optimum.
> [Figure of RBF kernel](https://anonymous.4open.science/r/nonmyopia/images/kernels/kernel_RBF_0.05_r-spotlight_init-1_hidden64_rs64.png)
> [Figure of Matern kernel](https://anonymous.4open.science/r/nonmyopia/images/kernels/kernel_Matern-1.5_0.05_r-spotlight_init-1_hidden64_rs64.png)
>
> *3. Hyperparameter Configurations for Baselines:* We explored various settings for the $\beta$ parameter in UCB, which governs the trade-off between exploration and exploitation. The findings indicate that despite tuning, UCB frequently becomes trapped in local optima due to its myopic nature. For other acquisition functions (SR, PI, KG, EI, MSL), hyperparameters directly influencing this balance are absent, so no additional experiments were conducted for these methods.
>
> [Visualization for UCB](https://anonymous.4open.science/r/nonmyopia/images/ucb/ucb_0.05_r-spotlight_init-1_hidden64_rs64.png)
>
> [Metrics for UCB](https://anonymous.4open.science/r/nonmyopia/images/ucb/ucb_final_yA.png) -- From the noon and in counter-clockwise: Ackley, Alpine and SynGP
>
> Thirdly, regarding your concerns about the integration of cost functions into other baselines, we provide the following clarifications. Our optimization objective is presented in line 188. In our experiments, we set $\lambda = 1$ for consistency, though in real-world applications, $\lambda$ is scenario-dependent, reflecting human preferences and contextual constraints. For a fair comparison with other methods, we incorporated cost constraints into their respective acquisition functions. Specifically, their optimization objectives of EI, PI, SR, and UCB are adapted to:
>
> \begin{equation}
> x = \arg \inf_{x \in \mathcal{X}} \left[ \mathbb{E} \_{p_t(f)} [\ell(f, x)] + \lambda c(x_{1:t}, x) \right],
> \end{equation}
>
> and that of KG is
> \begin{equation}
> x = \arg \inf_{x \in \mathcal{X}} \left[ \mathbb{E} \_{p_t(y | x)} [ \inf_{a \in \mathcal{A}} \{ \mathbb{E} \_{p \_{t+1}(f)} \ell(f, a)] + \lambda c(x_{1:t}, x, a) \} \right].
> \end{equation}
>
> This ensures that both myopic and non-myopic methods consider cost constraints in the optimization process. Our problem focuses on optimizing for the global maximum. Therefore, we use the final observed value in synthetic experiments and Bayesian regret in protein experiments as our evaluation metrics. Each experiment is given a fixed budget and will be terminated if the costs exceed the given budget.

---

> ### Author Response · Authors · 2024-11-23
> **Responses for minor points**
>
> Regarding the minor points related to the DAD paper, we appreciate the reviewer’s suggestions and have corrected the reference citation. We also clarify the distinctions between the DAD method [DAD] and our approach:
>
> *1. Dynamic Cost Consideration:* The DAD method does not account for dynamic costs, which limits its applicability to cost-constrained problems, such as protein design with restricted editing positions and amino acid choices. In contrast, our work specifically addresses non-myopic planning in dynamic cost settings, where the cost structure depends on specific problem. This capability is crucial for applications requiring adaptive optimization under such constraints.
>
> *2. Policy Network Adaptability:* In the DAD method, the policy network $\pi_\theta$ is pre-optimized and remains fixed during the design process. This static policy is a significant limitation in dynamic cost settings, where the cost for subsequent queries depends on earlier queries. In contrast, our method adapts the policy network iteratively, allowing for accurate and effective planning in scenarios with evolving cost constraints.
>
> Regarding concerns about our choice of surrogate models in synthetic settings, particularly the omission of Bayesian linear regression, it is important to note that this model is inherently limited to linear surface modeling. In our experiments, the synthetic settings involve a non-linear target function, making Bayesian linear regression unsuitable for accurately approximating the ground truth. To demonstrate this limitation, we compared the posterior surface generated by Bayesian linear regression with those of other kernel-based methods, as shown in the figure below. These results confirmed its inadequacy, leading us to exclude it from our ablation study.
>
> [Figure of posterior surfaces with different kernels](https://anonymous.4open.science/r/nonmyopia/images/kernels/posterior_surface.png)
>
> References:
> [DAD] Foster, Adam, et al. "Deep adaptive design: Amortizing sequential Bayesian experimental design." International conference on machine learning. PMLR, 2021.

---

> ### Author Response · Authors · 2024-11-23
> **Responses for question 1-3**
>
> Regarding your questions, we address them below:
>
> **Question 1:** The paper discusses the idea of "pathwise sampling" in lines 276-. Can the authors clarify how this approach of sampling a function from the probabilistic model and then optimizing a sequence of steps over this (now fixed) function is different from trajectory sampling [1] used in PETS, with similar ideas being used in model-based RL with PILCO [2]?
>
> **Answer 1:** We thank you for your question and the opportunity to clarify the idea of pathwise sampling. Regarding your concerns about the differences between the pathwise sampling approach discussed in the paper with trajectory sampling used in PETS and PILCO, the PE-TS paper presents an approach to modeling both aleatoric and epistemic uncertainty during planning. This method involves training multiple probabilistic neural networks using bootstrapped subsets of the observed data points. At each step, it initializes a number of particles (corresponding to the number of probabilistic neural networks) and assigns each particle to a probabilistic neural network for planning.
>
> * The PILCO paper uses trajectory sampling to address the issue of model bias in model-based reinforcement learning. Model-based methods often suffer from model bias, where the learned dynamics model may not accurately represent the real environment. By using trajectory sampling, PILCO incorporates model uncertainty into long-term planning, thereby reducing the impact of model bias.
>
> * Our method, however, leverages Thompson sampling, aiming to balance exploration and exploitation during planning. At each step, a posterior sample is drawn from the posterior distribution, as modeled by the surrogate probabilistic model, to guide the planning process. To improve efficiency, we adopt pathwise sampling [PathwiseSampling] for generating posterior samples, which circumvents the computational inefficiencies of traditional posterior sampling methods. By employing Thompson sampling in conjunction with pathwise sampling, our approach not only exploits knowledge from the current observed data but also preserves the exploratory capacity required for effective planning under uncertainty.
>
> **Question 2:** Why did the authors choose 64 restarts? Are the results meaningfully different with a different number of restarts? Requiring such a large number of restarts seems like a drawback of the proposed approach. Moreover, it would benefit the manuscript to explain what is meant by a "restart"; I assume it refers to starting the optimization process from multiple initial conditions.
>
> **Answer 2:** Thank you for the question. We use 64 restarts based on our computational budget. Restarting involves repeating the optimization process multiple times under the same conditions (i.e., initial points and surrogate model), which enhances robustness and ensures a more comprehensive exploration of the solution space. This technique is also employed in AlphaFold2 to identify the global optimal structure of proteins [AlphaFold_Multimer]. More restart generally leads to better performance, and we can afford 64 restarts in our experimental setting. In our implementation, we facilitate batching to handle the restarts. Specifically, with the current $x_t$, we expand it to a vector for a batch of 64 restarts and perform the same optimization procedure. This ensures robustness and increases the likelihood of finding the global optimum.
>
> **Question 3:** It is mentioned that the MSL method is not applicable to "real-world" tasks and referred to Section 4.2 (line 395), but I do not find any reference to MSL in the following.
>
>
> **Answer 3:** Thank you for the question. Typically, MSL assumes optimization directly on continuous input variables, which limits its applicability to real-world tasks such as protein design, where variables are discrete. Indeed, in Section 4.2, we elaborate on the protein design task, where the decision involves choosing which amino acid to mutate and which amino acid to mutate to. Even for a moderate-sized 300-token protein sequence and a myopic decision method, there are $300 \times 20$ options for a single decision. For a $T$-step lookahead, there are $(300 \times 20)^T$ discrete options, which is a scale at which many discrete optimizers required by MSL are likely to fail. Fortunately, these options exist in a semantic space, meaning that options similar to others are likely to have similar black-box outputs. Our approach exploits this by extending MSL with a variational network, allowing for efficient optimization in settings involving discrete variables.
>
>
> **References:**
>
> [AlphaFold_Multimer] Evans, Richard, et al. "Protein complex prediction with AlphaFold-Multimer. 2022; bioRxiv doi: 10 March 2022, https." doi. org/10.1101/2021.10 4.
>
> [PathwiseSampling] Wilson, James T., et al. "Pathwise conditioning of Gaussian processes." Journal of Machine Learning Research 22.105 (2021): 1-47.

---

> ### Author Response · Authors · 2024-11-23
> **Responses for questions about  experiments on protein design**
>
> **Question 4:** Given that predictions of an LLM are used as the "ground-truth" fluorescence levels, could the protein sequence with large edit distance (Figure 4) be outliers / wrong predictions of the LLM? Otherwise, what are alternative explanations for the S-shaped curve in Figure 4?
>
> **Answer 4:** We would like to clarify as follows. In this experiment, our goal is to demonstrate that our method can be applied to real-world applications, especially complex, semantically-rich tasks like protein design with various constraints while balancing our available computational budget. For details, we start with an arbitrary protein sequence and mutate it at 12 positions with 2 amino acids each, resulting in a protein space of 4096 proteins for this experiment. We assign the reward value for each protein as the fluorescence value. Since we do not involve wet-lab testing, to obtain the fluorescence value for these proteins, we build a simple Bayesian linear regression model on a large protein fluorescence dataset (ProteinEA, https://huggingface.co/datasets/proteinea/fluorescence) and use it as the oracle (i.e., the ground-truth reward function). Originally, the space constructed using the oracle is nearly convex, which means any optimization methods (including myopic and non-myopic ones) can achieve the global optimum. This, however, does not showcase the power of non-myopic methods. Thus, we add a synthetic function $g(x)$ to the ground-truth reward (the predicted fluorescence from the oracle) to achieve the S-shaped curve shown in Figure 4. The S-curve includes one local mode, which can act as a trap for myopic methods, as they are optimized to achieve immediate rewards. Our method, however, tolerates lower immediate rewards to escape the local mode and achieve the global optimum. Moreover, we also include more protein experiments with a different starting protein and a different S-curve (including 2 local modes) to demonstrate the robustness of our method. In these two additional experiments, our method with lookahead capacity outperformed the other ones.
>
> *Same starting protein, different $g(x)$ with 2 local modes*
>
> [Figure of distribution of fluorescence level](https://anonymous.4open.science/r/nonmyopia/images/additional_protein_exps/mutants_2p12_v1_v2_hop.png)\
> [Observed fluorescence level and Bayesian regret across BO steps](https://anonymous.4open.science/r/nonmyopia/images/additional_protein_exps/m1f2_yA_regret.png)
>
> *Different starting protein, same $g(x)$*
>
> [Figure of distribution of fluorescence level](https://anonymous.4open.science/r/nonmyopia/images/additional_protein_exps/mutants_2p12_v2_v1_hop.png)\
> [Observed fluorescence level and Bayesian regret across BO steps](https://anonymous.4open.science/r/nonmyopia/images/additional_protein_exps/m2f1_yA_regret.png)
>
>
> **Question 5:** How are the hyperparameters chosen? Are they chosen on the validation set over which is also optimized?
>
> **Answer 5:** Some hyperparameters in our experiment are the number of lookahead steps and re-generation times. They were selected to balance our computational budget and optimization effectiveness. For instance, although a larger number of lookahead steps could be chosen, generating longer sequences with an LLM is very costly due to its quadratic complexity. We set the number of BO steps to at least 12, as the starting protein needed at least 12 edits to reach the global maximum. We chose 16 steps to provide a buffer for potential errors during the complex optimization process with the LLM.
>
> **Question 6:** Why is the edit distance represented in the reward value (via g) and not only in the cost?
>
> **Answer 6** Thank you for the question. We realize our writing in the paper might have been confusing. We want to clarify that the function $g$ is chosen to be the oracle blackbox function, which depends on the edit distance to emulate the dynamic cost setting. By incorporating the edit distance into the reward value, $g$ effectively captures the relationship between modifications and their impact on the optimization process, providing a more realistic evaluation in a dynamic cost environment.
>
> **Question 7:** How is λ chosen?
>
> **Answer 7:** The $\lambda$ value is set to 1 in all of our synthetic experiments. In the protein experiment, the cost is spotlight cost and managed through re-generation, so we do not specify $\lambda$ in this context. In practice, the $\lambda$ value depends on various real-world factors and human preferences and should be determined on a case-by-case basis. For instance, when deciding on the best place to live, one might start by exploring nearby locations or opt to move to a distant place, depending on budget constraints and personal preferences.

---

> ### Author Response · Authors · 2024-11-23
> **Responses for questions about experiments on protein design (continue)**
>
> **Question 8:** How does the regret reflect the cost of the chosen protein? That is, in what sense does the cost function c affect the outcome of the empirical evaluation at all?
>
> **Answer 8:** In this experiment, we use the spotlight cost, which means that any edit not within the constraint incurs an infinite cost. We handle this cost by using regeneration, ensuring that all generated candidate proteins satisfy the constraint of having an edit distance equal to 1. As a result, the spotlight cost is effectively zero and does not impact the LLM optimization. Therefore, the reported regret solely indicates the quality of the generated proteins.
>
> **Question 9:** It is not clear to me how the authors optimize the variational objective with the non-Bayesian LLM in this setting. Can the authors clarify in more detail how this is related to the algorithm proposed in the main text?
>
> **Answer 9:** Thank you for your question. We use a variational network, i.e. a policy, and not a variational objective. The role of the neural network $\xi$ is to propose a new candidate: $\xi: (x \_{1:t}, y \_{1:t}) \mapsto x_{t+1}$, where $x_t$ represents the chosen protein at step $t$. Because the protein space is discrete, we cannot directly use equation (2) (line 256). Instead, we leverage the REINFORCE algorithm to optimize the neural network policy. We provide details of our objective function in line 269, where $\log p_t(x_{l+1} | x_{1:l}, \xi)$ is the log probability of the generated protein sequence from the neural network (which is differentiable), and $\text{EHIG} \_t(x \_{1:l+1})$ is the objective value computed by the surrogate model.
>
> **Question 10:** It is mentioned that steps sometimes "fail" (line 494). Can the authors elaborate what is meant by this and clarify how often it fails for each of the discussed baselines?
>
> **Answer 10:** The failure we mentioned in line 494 is that the LLM failed to generate a new sequence that satisfies the spotlight cost constraints (2 types of amino acids at 12 positions). For example, the LLM might be hallucinated or not follow the instructions so that it generates a sequence with a third amino acid (which is not in the allowed set) or it edits at the wrong position. In these cases, we re-generate the sequence again.  Using a more robust LLM (such as LLama with 70B parameters) might not face these issues. However, to balance our computational budget, or in real use cases where computation is limited, we use Llama-3.2 3B, thus we have to deal with these issues.
>
> **Question 11:** The paper provides no details on how the baselines are evaluated in this setting. Can the authors clarify this? In particular, can the authors clarify how "uncertainty" is measured in this context when we do not have a Bayesian model (as in the case of Llama-3.2). Or is a Bayesian linear regression uses, in which case I wonder how this can be compared to the Llama-3.2 policy network used in the proposed approach?
>
> **Answer 11:** The baseline methods used in this study are based on myopic objectives, and we experimented with various myopic baselines employing different heuristics, such as UCB and Expected Improvement (EI). In all baseline methods, we used a policy network since traditional discrete Bayesian optimization is not suitable for this setting. Specifically, we utilized Bayesian linear regression as a surrogate model, which takes the embedding of newly generated proteins to predict the distribution of fluorescence levels. The acquisition function then uses this distribution to compute the acquisition value. For example, UCB computes the mean and standard deviation from this distribution to obtain the acquisition value, which is subsequently used as the reward for optimizing the neural network policy using the REINFORCE algorithm, as previously explained. The Llama-3.2 3B serves as the neural network policy, generating the next candidate protein based on previous proteins and fluorescence values. By sampling from the output distributions of the neural network policy, we obtain different output proteins. If some proteins do not satisfy the constraints of edit distance, we perform sampling again (as explained above). You can find more about our implementation at this anonymized Github: https://anonymous.4open.science/r/nonmyopia

---

> > ### Comment · Reviewer_aZur · 2024-11-25
> >
> > I thank the authors for their detailed response to my questions and their efforts during the rebuttal.
> >
> > I still believe that some core limitations and weaknesses of evaluation remain unaddressed. Further, based on the other reviews, it seems to me that the presentation may be improved substantially by addressing some mentioned weaknesses and questions directly in the manuscript. I suspect that a major revision would be needed to address all concerns in their entirety. I am inclined to keep my current score.
> >
> > In the following, I include my comments on the main weaknesses that I outlined, and which I feel remain unaddressed.
> >
> > * I do not follow the authors argument about "receptive field" of non-myopic and myopic methods. I find this argument very handwavy and the manuscript would certainly benefit from a thorough theoretical analysis of this issue. The idea behind many myopic methods for BO is precisely to use "optimism" to ensure global optimization. Under some relatively weak assumptions, this can be guaranteed for methods like GP-UCB (see the Srinivas et al paper from 2009). I do not agree with the simplistic argument that "optimism" limits the "receptive field".
> > * I would like to mention that the output scale of the GP influences the "optimism" of *all* acquisition function; a larger output scale leading to more optimism.
> > * Further, I would like to refer the authors to the original paper on GP-UCB (and subsequent works) which derive bounds on $\beta$ for which GP-UCB is guaranteed to converge. Typically, for practical convergence $\beta$ has to be chosen much larger than $\beta=2$ as done in the ablations here.

---

> > > ### Author Response · Authors · 2024-11-27
> > > **Response for reviewer comment**
> > >
> > > Dear Reviewer aZur,
> > >
> > > Thank you for your thoughtful feedback. We appreciate your comment regarding the myopic methods, and we would like to provide additional clarification on the optimism within these approaches.
> > >
> > > First, our discussion on the receptive field was intended to illustrate the area in the input/output space that a decision-maker can consider when choosing the next query. The dynamic cost setting and budget constraint we consider, coupled with the myopic strategy, induced a narror receptive field, which limited what the decision maker can foresee beyond its immediate surroundings and can only make decisions based on such limited area. For example, in the spotlight cost setting that we consider, querying within an r-radius around the current point is free, but query anywhere outside of such spotlight is infinitely expensive. In this case, the budgeted dynamic cost setting makes it such that it is impossible to move very far away from the current location – if the decision maker tries to query point that are outside of the spotlight cost, the episode immediately terminates due to exhausting the query budget. All acquisitions, regardless of being myopic or not, need to respect this hard constraint induced by the the budgeted, dynamic cost BO setting that we are considering. We want to emphasize that, when facing with other cost settings, such as unbudgeted BO or BO with uniform cost, myopic strategy might not face the challenge of narrow receptive field, and strategy like UCB might potentially work very well.
> > >
> > > We understand that using the UCB acquisition function, the level of optimism can be controlled by the $\beta$ hyperparameter. A smaller $\beta$ prioritizes exploitation, while a larger $\beta$ prioritizes exploration. With sufficiently large $\beta$, the standard deviation term dominates the mean term, leading to decisions driven by the most uncertain areas. To further illustrate the impact of large $\beta$, we conducted additional experiments with $\beta$ values ranging from 0.1 to 1000 on 9 synthetic functions commonly used in BO literature. In the figure below, we highlight UCB’s behavior with increasing $\beta$.
> > >
> > > [Comparison of ours and UCB with various $\beta$](https://anonymous.4open.science/r/nonmyopia/images/ucb/ucb_0.05_r-spotlight_init-1_hidden64_rs64.png)

---

> > > ### Author Response · Authors · 2024-11-27
> > > **Response for reviewer comment**
> > >
> > > We also provide the value of the final action, normalized to range from -1 to 1, where -1 represents the worst outcome and 1 the best.
> > >
> > > | (Environment, Noise level, Cost Function) | Ours      | UCB ($\beta = 0.1$)   | UCB ($\beta = 0.5$)   | UCB ($\beta = 1$)   | UCB ($\beta = 2$)   | UCB ($\beta = 5$)   | UCB ($\beta = 10$)   | UCB ($\beta = 20$)   | UCB ($\beta = 50$)   | UCB ($\beta = 100$)   | UCB ($\beta = 200$)   | UCB ($\beta = 500$)   | UCB ($\beta = 1000$)   |
> > > |:-----------------------------------------------|:----------|:----------------------|:----------------------|:--------------------|:--------------------|:--------------------|:---------------------|:---------------------|:---------------------|:----------------------|:----------------------|:----------------------|:-----------------------|
> > > | (Ackley, 0.05, r-spotlight)         | 0.97±0.03 | 0.7±0.4               | 0.4±0.41              | 0.4±0.41            | 0.4±0.41            | 0.7±0.43            | 1.0±0.01            | 0.71±0.35            | 0.55±0.65            | 0.45±0.41             | -0.34±0.38            | -0.12±0.13            | 0.06±0.5               |
> > > | (Ackley4D, 0.05, r-spotlight)       | 0.97±0.02 | 0.68±0.44             | 0.68±0.44             | 0.67±0.43           | 0.68±0.44           | 0.98±0.01           | 0.98±0.01           | 0.87±0.17            | 0.99±0.01           | 0.47±0.33             | -0.72±0.06            | -0.33±0.3             | -0.6±0.24              |
> > > | (Alpine, 0.05, r-spotlight)         | 0.99±0.0  | -0.01±0.01            | -0.01±0.01            | -0.01±0.01          | -0.01±0.01          | -0.01±0.01          | -0.01±0.01          | 0.15±0.24            | 0.73±0.36            | 0.92±0.09             | 0.15±0.46             | -0.43±0.48            | -0.2±0.66              |
> > > | (Cosine8, 0.05, r-spotlight)        | 0.93±0.01 | 0.96±0.01             | 0.97±0.01             | 0.96±0.02           | 0.97±0.02           | 0.96±0.03           | 0.97±0.02           | 0.97±0.04            | 0.88±0.04            | 0.71±0.06             | 0.62±0.07             | 0.13±0.18             | -0.18±0.1              |
> > > | (Hartmann, 0.05, r-spotlight)       | 0.96±0.03 | 0.95±0.02             | 0.94±0.04             | 0.95±0.03           | 0.95±0.03           | 0.97±0.02           | 0.96±0.03           | 0.97±0.03            | 0.94±0.03            | 0.95±0.03             | 0.81±0.12             | -0.02±0.61            | -0.52±0.43             |
> > > | (HolderTable, 0.05, r-spotlight)    | 0.05±0.08 | -0.49±0.01            | -0.5±0.01             | -0.49±0.01          | -0.49±0.01          | -0.49±0.01          | -0.28±0.32          | -0.03±0.34           | 0.81±0.6            | 0.9±0.81              | 0.76±0.23             | 0.27±1.1              | -0.41±0.32             |
> > > | (Levy, 0.05, r-spotlight)           | 0.95±0.0  | 0.87±0.0              | 0.87±0.0              | 0.87±0.0            | 0.87±0.0            | 0.87±0.0            | 0.87±0.0            | 0.87±0.0             | 0.87±0.0            | 0.86±0.01             | 0.93±0.05             | 0.82±0.12             | 0.84±0.06              |
> > > | (StyblinskiTang, 0.05, r-spotlight) | 1.0±0.0   | 0.91±0.0              | 0.91±0.0              | 0.91±0.0            | 0.91±0.0            | 0.91±0.0            | 0.91±0.0            | 0.91±0.0             | 0.91±0.0            | 0.94±0.04             | 0.93±0.1              | 0.96±0.06             | 0.92±0.07              |
> > > | (SynGP, 0.05, r-spotlight)          | 0.63±0.25 | 0.45±0.01             | 0.45±0.01             | 0.45±0.01           | 0.45±0.01           | 0.45±0.01           | 0.45±0.01           | 0.45±0.01            | 0.45±0.01           | 0.45±0.01             | 0.22±0.32             | 0.61±0.53             | 0.5±0.53               |
> > >
> > > These empirical results further illustrate that the large $\beta$ value can encourage the decision maker to make queries that highly prioritize exploration. As illustrated in the above figure and table, such exploration is typically myopic and unplanned, and consequently, the decision maker typically missed the global optima or overexplored the unpromising region. We also want to note that in our experiment, no single $\beta$ outperformed others in all settings; for example, $\beta = 10$ works well for Ackley but does not work for other functions. Indeed, choosing the value of $\beta$ for UCB prior to running the online experiment is nontrivial in practice.

---

> > > ### Author Response · Authors · 2024-11-27
> > > **Response for reviewer comment**
> > >
> > > Regarding our method, theoretically, we follow Bayes optimal decision rule from decision theory. Bayes optimal decision rules are used to make decisions under uncertainty by minimizing expected loss, which is the expected loss based on posterior probabilities of the unknown variable (in this case, the unknown future query). Non-myopic approaches adhering to Bayes optimal decision rules have consistently demonstrated greater effectiveness in prior research [1-3]. The naive implementation of Bayes optimal decision rule has an exponentially large number of decision variables, which is typically impractical to solve in practice, and this is precisely what we aim to address in this paper with a variational neural network. Our method reduces the number of decision variables to a constant with respect to the lookahead horizon length, ensure computational efficiency while maintaining the Bayes optimality. Empirically, our nonmyopic method consistently performs well across all environments with no additional hyperparameter (other than typical GP hyperparameters such as lengthscale or choice of kernel).
> > >
> > > Thank you again for your valuable feedback and for engaging with our work. We hope this response provides clarity for your questions, and please let us know if you have any further concerns. We look forward to hearing from you.
> > >
> > > [1] Peter Frazier, Warren Powell, and Savas Dayanik. The knowledge-gradient policy for correlated
> > > normal beliefs. INFORMS journal on Computing, 21(4):599–613, 2009.
> > >
> > > [2] Shali Jiang, Daniel Jiang, Maximilian Balandat, Brian Karrer, Jacob Gardner, and Roman Garnett.
> > > Efficient nonmyopic Bayesian optimization via one-shot multi-step trees. Advances in Neural
> > > Information Processing Systems, 33:18039–18049, 2020b.
> > >
> > > [3] Liu, Peng, Haowei Wang, and Wei Qiyu. "Bayesian Optimization with Switching Cost: Regret Analysis and Lookahead Variants." IJCAI. 2023.

---

### Official Review · Reviewer_h4kf · 2024-11-02

**Soundness:** 3
**Presentation:** 3
**Contribution:** 2
**Rating:** 5
**Confidence:** 3

**Summary:**

In this paper, the authors study nonmyopic Bayesian optimization in dynamic cost settings, where costs can exhibit both Markovian and non-Markovian properties. The authors propose to use recurrent neural networks for variational lookahead optimization which can greatly increase the lookahead horizon beyond existing works. Empirical evaluations on synthetic test functions and a protein design task demonstrate the effectiveness of the proposed method.

**Strengths:**

1. This paper systematically studies many classes of varying cost structures. This contribution is novel.

2. The proposal of using RNNs for lookahead optimization is novel and the effect of extending the lookahead horizon seems significant.

3. The experiment results seem promising in demonstrating the positive effect of having long lookahead horizons.

**Weaknesses:**

1. There is only one lookahead baseline in the empirical evaluation. The authors mentioned several others in the related works section [1, 2, 3]. Can the authors explain why they were not included in the comparison? There is also [4] which is related to the current work.

2. A major claimed contribution is that extending lookahead horizon improves optimization results. This makes intuitive sense but there is no ablation study to confirm this claim. It would be helpful to compare different lookahead horizons’s impact on the final optimization results.

[1]: Lee, Eric Hans, et al. "A nonmyopic approach to cost-constrained Bayesian optimization." Uncertainty in Artificial Intelligence. PMLR, 2021.

[2]: Astudillo, Raul, et al. "Multi-step budgeted bayesian optimization with unknown evaluation costs." Advances in Neural Information Processing Systems 34 (2021): 20197-20209.

[3]: Wu, Jian, and Peter Frazier. "Practical two-step lookahead Bayesian optimization." Advances in neural information processing systems 32 (2019).

[4]: Liu, Peng, Haowei Wang, and Wei Qiyu. "Bayesian Optimization with Switching Cost: Regret Analysis and Lookahead Variants." IJCAI. 2023.

**Questions:**

1. Around line 100, the authors mentioned the synthetic functions “requiring 10 to 20 planning steps to find the global optimum”. How does one determine this number for a function?

2. In the protein sequence experiment, how did the authors arrive at the specific form for $g(x)$ (line 459)?

3. Can the authors provide more details regarding using Llama-3.2 3B as the variational network (line 483)?

---

> ### Author Response · Authors · 2024-11-24
>
> Dear Reviewer h4kf,
>
> Thank you for your valuable feedback. We would like to address the weaknesses and your questions below.
>
> First, regarding the inclusion of additional lookahead baselines mentioned in [1, 2, 3], we are currently working on analyzing these methods and will add more details later.
>
> Regarding [4], the primary difference between our approach and the one proposed in [4] lies in the integration of pathwise sampling and a variational network. While [4] introduces a cost-constrained lookahead acquisition function similar to ours, it does not incorporate pathwise sampling. Without pathwise sampling, the number of samples at each step grows exponentially. Additionally, without the variational network, the number of optimizing parameters also grows exponentially, which is intractable in settings requiring a large lookahead horizon. By integrating pathwise sampling and a variational network, our method significantly reduces computational complexity. This allows us to optimize language models for more complex practical tasks, addressing limitations that prior nonmyopic methods, including the one in [4], still face. We will include these analyses in the rebuttal revision.
>
> Secondly, regarding the analysis of lookahead effects, we have included experimental results on the ablation of the number of lookahead steps. Specifically, we experimented with the MSL baseline and our method using 5-, 10-, and 15-step lookahead on three synthetic functions: Ackley, Alpine, and SynGP. These results illustrate the relationship between the number of lookahead steps and the robustness of the optimization, providing insights into how the performance of our approach varies with different horizon lengths. Notably, with a larger lookahead horizon, the chance of finding the global optimum increases.
>
> [Figure of ablation study on lookahead horizon](https://anonymous.4open.science/r/nonmyopia/images/lookahead/lookahead_0.05_r-spotlight_init-1_hidden64_rs64.png)

---

> ### Author Response · Authors · 2024-11-24
> **Responses for questions**
>
> Regarding your questions, we address them below:
>
> **Questions 1:** Around line 100, the authors mentioned the synthetic functions “requiring 10 to 20 planning steps to find the global optimum”. How does one determine this number for a function?
>
> **Answer 1:** In synthetic settings, we estimate the number of planning steps based on the spotlight radius. Specifically, we compute the distance between the initial point and the maximum point, then divide this distance by the spotlight radius.
> In practical use cases, with a good prior, we can estimate the number of planning steps in a similar manner. For example, in the case of delivering packages without knowing the exact map, a delivery man can estimate the distance between their current location and the target location based on their prior knowledge (e.g., having traveled to nearby locations before).
>
> **Questions 2:** In the protein sequence experiment, how did the authors arrive at the specific form for g(x) (line 459)?
>
> **Answer 2:** We design the function $g(x)$ by hand, aiming to find a non-convex function with one local mode to demonstrate the benefits of lookahead optimization. Specifically, myopic methods might have some information about the global mode, but their acquisition functions are optimized to receive the highest reward at the next step. This limitation prevents them from escaping the local mode. In contrast, our method, when it has signals of the global mode, can tolerate lower rewards in intermediate steps to achieve a higher reward in the end. Additionally, we have included more protein experiments with a different starting protein and a different S-curve (including 2 local modes) to demonstrate the robustness of our method. In these two additional experiments, our method with lookahead capacity outperformed the other ones.
>
> *Same starting protein, different $g(x)$ with 2 local modes*
>
> [Figure of distribution of fluorescence level](https://anonymous.4open.science/r/nonmyopia/images/additional_protein_exps/mutants_2p12_v1_v2_hop.png)\
> [Observed fluorescence level and Bayesian regret across BO steps](https://anonymous.4open.science/r/nonmyopia/images/additional_protein_exps/m1f2_yA_regret.png)
>
> *Different starting protein, same $g(x)$*
>
> [Figure of distribution of fluorescence level](https://anonymous.4open.science/r/nonmyopia/images/additional_protein_exps/mutants_2p12_v2_v1_hop.png)\
> [Observed fluorescence level and Bayesian regret across BO steps](https://anonymous.4open.science/r/nonmyopia/images/additional_protein_exps/m2f1_yA_regret.png)
>
>
> **Questions 3:** Can the authors provide more details regarding using Llama-3.2 3B as the variational network (line 483)?
>
> **Answer 3:** We employ Llama-3.2 3B as the variational network, conceptualizing the optimization process as a dialogue between the protein designer and the LLM. In this setup, the protein designer provides the Llama model with protein(s) and corresponding reward(s) (e.g., fluorescence levels) in the previous time step(s) and requests it to modify the latest protein to enhance its properties. We chose Llama-3.2 3B due to its robustness in following instructions and our limited computational budget. Its smaller size allows us to perform experiments more efficiently, and its ability to follow instructions carefully makes it easier to add constraints, such as limiting the number of possible amino acids.
> To acclimate the Llama model to the protein editing task, we conducted additional supervised fine-tuning before initiating the optimization process. Further details about this experiment are available in Appendix B. More information about our implementation can be found at this anonymized GitHub repository: anonymous.4open.science/r/nonmyopia.

---

> ### Author Response · Authors · 2024-11-24
> **Response for weakness 1 regarding benchmarking the related works**
>
> Thank you for your feedback on the related works concerning non-myopic Bayesian optimization methods for dynamic cost settings. We appreciate the opportunity to clarify the relation between our work and the previous ones in terms of the cost structure. To facilitate the discussion, we survey the prior literature on the topic broadly and come up with a taxonomy based on two factors of the cost function: uncertainty and variability.
>
> - In terms of uncertainty, costs can be classified as known or unknown prior to making a decision. When the cost is unknown, it can be viewed as a random variable that can be modeled probabilistically.
>
> - In terms of variability, the cost structure can be categorized into dynamic costs, which vary based on the query history, and static costs, which remain fixed for a particular query over time. We note that when the cost is static, the BO community also studies two variations of this structure: heterogeneous cost and homogeneous cost. Homogeneous cost is the setting where the cost of all queries is the same, whereas heterogeneous cost is the setting where the cost of a query is a function of the query itself.
>
> Using this taxonomy, we classify the literature into four categories, as shown in the table below.
> |               | Known Cost                                                                                      | Unknown Cost                                                                                     |
> |---------------|-------------------------------------------------------------------------------------------------|--------------------------------------------------------------------------------------------------|
> | **Static Cost** | Do not vary based on the previous queries and predictable costs, easy to budget over time. Related literature: [4], [5], and [6]  | Do not vary based on the previous queries, but the actual amount is not fully known due to external factors.                                              Related literature: [1], [2], [3], and [7]   |
> | **Dynamic Cost** | Varies based on the previous queries, but can be quantified or predicted.                      Related literature: Ours | Varies based on the previous queries, and is difficult to predict precisely.                      Related literature: To the best of our knowledge, we have not yet encountered related work in this category, and we plan to work on this setting in our future direction. |
>
> To further illustrate the distinction between cost structures, we visualize the uncertainty and variability of these structures as probabilistic graphical diagrams below. In these diagrams, $f$ represents the target black-box function, $x$ denotes the input query, $y$ is the output value, and $c$ is the cost incurred by querying $x$. On the left — the dynamic-known cost structure — the cost of querying $x_3$ can depend on $x_1$ and $x_2$. On the right — the static-unknown cost structure — the cost of querying $x_3$ is independent of other queries.
>
> [Comparison of cost structures](https://anonymous.4open.science/r/nonmyopia/images/cost_structures.png)
>
>
> **References:**
>
> [1] Astudillo, R., Jiang, D., Balandat, M., Bakshy, E., & Frazier, P. (2021). Multi-step budgeted bayesian optimization with unknown evaluation costs. Advances in Neural Information Processing Systems, 34, 20197-20209.
>
> [2] Lee, E. H., Eriksson, D., Perrone, V., & Seeger, M. (2021, December). A nonmyopic approach to cost-constrained Bayesian optimization. In Uncertainty in Artificial Intelligence (pp. 568-577). PMLR.
>
> [3] Belakaria, S., Doppa, J. R., Fusi, N., & Sheth, R. (2023, April). Bayesian optimization over iterative learners with structured responses: A budget-aware planning approach. In International Conference on Artificial Intelligence and Statistics (pp. 9076-9093). PMLR.
>
> [4] Jian Wu and Peter Frazier. Practical two-step lookahead Bayesian optimization. Advances in neural information processing systems, 32, 2019
>
> [5] Nyikosa, F. M., Osborne, M. A., & Roberts, S. J. (2018). Bayesian optimization for dynamic problems. arXiv preprint arXiv:1803.03432.
>
> [6] Lam, R., Willcox, K., & Wolpert, D. H. (2016). Bayesian optimization with a finite budget: An approximate dynamic programming approach. Advances in Neural Information Processing Systems, 29.
>
> [7] Luong, P., Nguyen, D., Gupta, S., Rana, S., & Venkatesh, S. (2021). Adaptive cost-aware Bayesian optimization. Knowledge-Based Systems, 232, 107481.

---

> ### Author Response · Authors · 2024-11-24
> **Response for weakness 1 regarding benchmarking the related works (continue)**
>
> Our problem setting focuses on optimizing within a *known and dynamic* cost setting, which is an important cost structure in many practical applications as we motivated earlier. Previous literature has developed methods for complementary-but-distinct cost settings, and we believe that those methods are not suitable for the setting studied in this paper. We elaborate on our position further below by analyzing the cost structure of the suggested papers:
> - The papers *“Multi-step budgeted Bayesian optimization with unknown evaluation costs”*, *“A nonmyopic approach to cost-constrained Bayesian optimization”*, and *“Adaptive cost-aware Bayesian optimization”* address cost structures characterized by unknown, heterogeneous costs. For instance, in the hyperparameter optimization (HPO) problem studied in these papers, the cost of evaluating a hyperparameter set (i.e., training the target model with that set) is unknown but static for a particular model and a set of hyperparameters and does not depend on previously chosen sets.
> - The paper *“Bayesian optimization over iterative learners with structured responses: A budget-aware planning approach”* also incorporates an unknown, heterogeneous cost structure. The key distinction from the previously mentioned works is the inclusion of an additional cost factor: the number of training epochs for evaluating a hyperparameter set. In this context, for a given location \( x \) (a set of hyperparameters) and a specific number of training epochs, the cost is unknown but fixed, as it does not depend on prior queries (previously chosen sets of hyperparameters) or prior choices of training epochs.
>
> - Finally, the papers *“Practical two-step lookahead Bayesian optimization”*, *“Bayesian optimization for dynamic problems”*, and *“Bayesian optimization with a finite budget: An approximate dynamic programming approach”* focus on Bayesian optimization without accounting for cost structures. These works implicitly assume that all locations in the search space have the same, constant, and known cost. This implies that the cost is either zero or any fixed constant and, therefore, not subject to optimization.
>
> Based on the above evidence, the settings in these works are fundamentally different from ours.
>
> Regarding the non-myopic methods presented in the suggested papers [1–4], these approaches extend the Expected Improvement acquisition function to address non-myopic optimization challenges. While they provide valuable insights, these methods directly optimize free variables in a multi-step tree (MST), which introduces an exponential increase in the number of optimization variables as the lookahead horizon grows. In our context, incorporating their lookahead mechanisms would amount to combining a multi-step tree structure with a dynamic cost function, an approach that is already benchmarked in our work. The optimization objectives of these lookahead methods, once adapted, align with Equation (1) (lines 188–190) in our paper. Our planning algorithm distinguishes itself from the literature by integrating a policy neural network. In the context of this paper on the dynamic and known cost setting, we believe that our experiments have sufficiently benchmarked the previous planning strategy presented in the suggested papers. We are eager to hear the reviewer's thoughts on this to further improve our work and welcome any specific suggestions from the reviewer on how to incorporate the related nonmyopic BO methods as additional baselines for our setting. We again thank you for your thoughtful questions and we look forward to hearing more from you.
>
> **References:**
> [1] Astudillo, R., Jiang, D., Balandat, M., Bakshy, E., & Frazier, P. (2021). Multi-step budgeted bayesian optimization with unknown evaluation costs. Advances in Neural Information Processing Systems, 34, 20197-20209.
>
> [2] Lee, E. H., Eriksson, D., Perrone, V., & Seeger, M. (2021, December). A nonmyopic approach to cost-constrained Bayesian optimization. In Uncertainty in Artificial Intelligence (pp. 568-577). PMLR.
>
> [3] Belakaria, S., Doppa, J. R., Fusi, N., & Sheth, R. (2023, April). Bayesian optimization over iterative learners with structured responses: A budget-aware planning approach. In International Conference on Artificial Intelligence and Statistics (pp. 9076-9093). PMLR.
>
> [4] Jian Wu and Peter Frazier. Practical two-step lookahead Bayesian optimization. Advances in neural information processing systems, 32, 2019
>
> [5] Nyikosa, F. M., Osborne, M. A., & Roberts, S. J. (2018). Bayesian optimization for dynamic problems. arXiv preprint arXiv:1803.03432.
>
> [6] Lam, R., Willcox, K., & Wolpert, D. H. (2016). Bayesian optimization with a finite budget: An approximate dynamic programming approach. Advances in Neural Information Processing Systems, 29.
>
> [7] Luong, P., Nguyen, D., Gupta, S., Rana, S., & Venkatesh, S. (2021). Adaptive cost-aware Bayesian optimization. Knowledge-Based Systems, 232, 107481.

---

### Official Review · Reviewer_5XvL · 2024-11-03

**Soundness:** 3
**Presentation:** 2
**Contribution:** 3
**Rating:** 6
**Confidence:** 3

**Summary:**

This paper proposes a nonmyopic Bayesian optimization method to tackle the black-box optimization problem where the evaluation cost is action-dependent. The authors propose to optimize recurrent neural networks to choose the next sample position, and use pathwise sampling to reduce the exponential complexity during multi-step planning. The experiment results demonstrate that the proposed nonmyopic method is capable of locate global optimal region in synthetic functions, and the real experiment results over protein design demonstrate the practicality of the proposed method.

**Strengths:**

1. I think the problem tackled by this paper is novel, which corresponds to many real-world applications where traditional BO cannot be directly applied in.

2. The overall algorithm design is intuitive and clear.

3. The experiment results of protein design is impressive.

**Weaknesses:**

1. There is only one real world experiment with discrete action space. It would be better if evaluating the proposed algorithm in real-world/simulation tasks with continuous space.

2. There is no comparison with the methods that tackle optimization problems with dynamic cost, as metioned in the related work.

**Questions:**

1. What is the $\lambda$ value assigned in the experiment? How to define this value in practice?

2. In protein design experiment, the optional actions are restricted to 2 amino acid in each edited position. Why not use all 4 amino acid as candidate actions?

---

> ### Author Response · Authors · 2024-11-24
> **Response for weaknesses**
>
> Dear Reviewer 5XvL,
>
> Thank you for your insightful feedback. In response to the weaknesses highlighted, we have performed more experiments and added more detailed descriptions. First, one of the major concerns was the limited scope of real-world experiments. To address this, we have included additional results that expand on our previous work with protein design experiments. Specifically, we conducted experiments using a different starting protein and a distinct synthetic function (including two local modes) to demonstrate the flexibility, applicability, and robustness of our approach across various scenarios. In these two additional experiments, our method with lookahead capacity outperformed the others.
>
> *Same starting protein, different $g(x)$ with 2 local modes*
>
> [Figure of distribution of fluorescence level](https://anonymous.4open.science/r/nonmyopia/images/additional_protein_exps/mutants_2p12_v1_v2_hop.png)\
> [Observed fluorescence level and Bayesian regret across BO steps](https://anonymous.4open.science/r/nonmyopia/images/additional_protein_exps/m1f2_yA_regret.png)
>
> *Different starting protein, same $g(x)$*
>
> [Figure of distribution of fluorescence level](https://anonymous.4open.science/r/nonmyopia/images/additional_protein_exps/mutants_2p12_v2_v1_hop.png)\
> [Observed fluorescence level and Bayesian regret across BO steps](https://anonymous.4open.science/r/nonmyopia/images/additional_protein_exps/m2f1_yA_regret.png)
>
> Moreover, we are working on human travel optimization problems using maps captured from NASA's Earth Observatory (https://earthobservatory.nasa.gov/features/NightLights). This problem involves optimizing in a human travel space, modeled as a 2D continuous domain, providing a realistic and complex scenario with a continuous action space.
> Another concern raised by the reviewers was the lack of comparison with existing methods that address optimization problems with dynamic costs. In response, we have incorporated dynamic cost constraints into all our baselines (SR, EI, PI, UCB, KG, MSL) for comparison. To ensure a fair and comprehensive comparison, we modified these methods to integrate cost constraints directly into their respective acquisition functions. Specifically, their optimization objectives of EI, PI, SR, and UCB are adapted to:
>
> \begin{equation}
> x = \arg \inf_{x \in \mathcal{X}} \left[ \mathbb{E} \_{p_t(f)} [\ell(f, x)] + \lambda c(x_{1:t}, x) \right],
> \end{equation}
> and that of KG is
> \begin{equation}
> x = \arg \inf_{x \in \mathcal{X}} \left[ \mathbb{E} \_{p_t(y | x)} [ \inf_{a \in \mathcal{A}} \{ \mathbb{E} \_{p \_{t+1}(f)} \ell(f, a)] + \lambda c(x_{1:t}, x, a) \} \right].
> \end{equation}
>
> Using these objective functions, we ensured that all methods are under the same conditions, thus demonstrating the effectiveness of our method in terms of lookahead capacity. The baseline results in our paper are already derived from the modified methods as presented above.

---

> ### Author Response · Authors · 2024-11-24
> **Responses for questions**
>
> Regarding your questions, we address them below:
>
> **Questions 1:** What is the λ value assigned in the experiment? How to define this value in practice?
>
> **Answer 1:** Currently, the $\lambda$ value is set to 1 in all of our synthetic experiments. In practice, the $\lambda$ value depends on various real-world factors and human preferences and should be determined on a case-by-case basis. For instance, when deciding on the best place to live, one might start by exploring nearby locations or opt to move to a distant place, depending on budget constraints and personal preferences.
> In the protein design experiment, we use the spotlight cost, which means that any edit not within the constraint incurs an infinite cost. We handle this cost by using regeneration (regenerate a new candidate protein if the current one fails to meet the spotlight cost constraint, line 493-496), ensuring that all generated candidate proteins satisfy the constraint of having an edit distance equal to 1. Thus, we do not specify the $\lambda$ for this experiment.
>
> **Question 2:** In protein design experiment, the optional actions are restricted to 2 amino acid in each edited position. Why not use all 4 amino acid as candidate actions?
>
> **Answer 2:** Currently, to balance between demonstrating the robustness of our method and our computational budget, we restrict the actions to 2 amino acids and 12 positions. This setup ensures that there is only one possible protein with the highest reward after 12 edits, imposing stricter constraints and requiring the method to be robust enough to find the global optimal solution. Allowing 4 amino acids per position could result in multiple proteins with the highest reward, making the problem easier. Moreover, our setting is closer to practical situations where amino acids are often replaced with those in the same group (e.g., having the same negative electric charge) to maintain protein functionality. We also performed two more protein design experiments with a different starting protein and a different synthetic function (with 2 local modes) as presented above.
> You can find more about our implementation at this anonymized Github: https://anonymous.4open.science/r/nonmyopia

---

> > ### Comment · Reviewer_5XvL · 2024-11-24
> >
> > Thanks for your clarification and additional experiments over the protein design problems, and I look forward to the experimental results on the human travel optimization problems. I think the proposed algorithm has demonstrated its practicality in real-world applications. However, I agree with the Reviewer xEWL and h4kf that cost-aware and budget-constrained nonmyopic BO approaches mentioned in the related work should be also included in the baseline, instead of using modified version traditional BO methods. I will maintain my current positive score.

---

> > > ### Author Response · Authors · 2024-12-01
> > > **Response for additional real-world experiments**
> > >
> > > Dear Reviewer 5XvL,
> > >
> > > We conducted additional experiments on human travel optimization in a 2D continuous domain. Specifically, we utilized an image from NASA's Earth Observatory\footnote{https://earthobservatory.nasa.gov/features/NightLights}, which is a 2016 grayscale image of night lights in Georgia and South Carolina states, with a resolution of $1000 \times 1000$ pixels. To facilitate the optimization of the GP surrogate model and avoid numerical issues due to image noise, we applied a stack blur with a radius of 40 to the image. The pixel values, ranging from 0 to 255, were normalized to a range of $-3$ to $3$. The image width and height were normalized to a range of $0$ to $1$. We apply our methods and baselines with spotlight cost ($r=0.1$) and Euclidean cost. The kernel for GP surrogate model is RBF.
> > >
> > > Below figures show the results of our methods and baselines on spotlight and Euclidean cost, respectively. In this environment, nonmyopic methods demonstrated their advantage in lookahead capability. Notably, our method showed its effectiveness in directly reaching the global optimum rather than querying around sub-optimal locations before approaching the global optimum as MSL.
> > >
> > > [Results on NightLight experiment with Euclidean cost](https://anonymous.4open.science/r/nonmyopia/images/nightlight/nightlight_0.0_euclidean_init-1_hidden64_rs64.png)
> > >
> > > [Results on NightLight experiment with spotlight cost](https://anonymous.4open.science/r/nonmyopia/images/nightlight/nightlight_0.0_r-spotlight_init-1_hidden64_rs64.png)

---

> > > > ### Comment · Reviewer_5XvL · 2024-12-01
> > > >
> > > > Thank you for your additional experimental results.

---

### Official Review · Reviewer_xEWL · 2024-11-03

**Soundness:** 2
**Presentation:** 3
**Contribution:** 2
**Rating:** 3
**Confidence:** 5

**Summary:**

The classical Bayesian Optimization (BO) frameworks operate under the assumption of static query costs. However, in practical scenarios, dynamic cost structures often arise. Hence, query costs dynamically change with respect to the previous queries. To address this, the paper proposes a cost-constrained nonmyopic BO algorithm that incorporates dynamic cost models into the BO framework. The method employs pathwise Thompson sampling and variational optimization. Empirical evaluation on both synthetic and real-world problem domains shows the effectiveness and scalability of the proposed approach against myopic baselines.

**Strengths:**

- The proposed nonmyopic BO approach handles dynamic cost structures that are inherent to many real-world problem contexts.
- The empirical evaluation covers both synthetic and real-world problem domains, which demonstrates the applicability of the proposed approach in real-world problems.
- A complexity analysis for approximating the posterior predictive distribution via path-wise sampling is provided.
- The empirical analysis is presented considering different cost functions.
- Using an LLM as a variational network of the proposed approach and showcasing the application of the proposed method on the protein design process as a dialogue is interesting.

**Weaknesses:**

- The empirical evaluation is not supportive enough to show the effectiveness of the proposed nonmyopic approach mainly due to the following reasons:
1. In Section 2, the paper cites many nonmyopic BO approaches from the literature, e.g. [1,2,3,4], including cost-aware and budget-constrained ones. However, in empirical analysis, these methods are not chosen as baselines, hence the empirical evidence is not strong enough to support contribution. It would have been supported better if the proposed method was compared against some of such approaches, which would clarify the contribution of using (pathwise Thompson sampling & variational network)-based proposed idea against existing lookahead ideas. Further, the method could have been empirically compared against those methods in terms of computational efficiency, since enhancing the computational efficiency of nonmyopic BO is one of the claims of the paper (line 120).

2. It is stated that the proposed approach can be more robust when the approximation of the reward function changes significantly between queries. However, it could have demonstrated the robustness of the approach better, if the behavior of the approach with respect to the approximate reward function throughout active learning queries.

3. The paper concentrates on the HES acquisition function for the proposed approach. However, if the proposed approach is flexible and generalizable, then it could have shown its flexibility better if any other acquisition function is considered, such as knowledge gradient and expected improvement. It would also directly show the change in the performance among myopic and nonmyopic BO versions under the same acquisition function.

4. There are many missing important ablation analyses, which would have supported the effectiveness of the proposed approach better with respect to its individual parts, specifically:

- 5. The performance under different look ahead step sizes. How does the performance of the proposed approach change under a short look-ahead horizon (budget)? It would support the empirical performance with respect to different horizon lengths.

- 6. Regarding the initial surrogate model, an ablation on the quality of the initial surrogate GP with respect to (1) initial training set size (which is not stated in the paper?), and (2) kernel choice other than Matern would strengthen the empirical analysis.

- 7. Although a real-world problem (protein sequence design) is chosen as the second test bed, the analysis is not detailed and scalable considering the settings. That is, an ablation study regarding the different $T$ (number of edits) values would have shown the applicability of the proposed approach better. Because most protein sequences have a sequence search space >> 4096, such as a typical human protein is of length 200-300 which makes the search space enormous. There are many generative and BO-based approaches that can design considering all positions of the sequence efficiently, hence considering a very limited design setting does not support the applicability and scalability of the proposed approach well enough.

- 8. Similar to the above point, why only 2 amino acid types is considered for the design, among possible 20 choices? How does the empirical performance and computational efficiency of the method change with respect to increased search space?

- 9. Batched BO is a conventional method also demonstrated to be effective in biological sequence design problem. An ablation study on the increased batch size would help for a better analysis.

- 10. Lastly, an ablation analysis with respect to the variational network (e.g. using a variational network of different accuracy levels) on a problem domain would support the actual benefit of the proposed approach.

11. It would have been supported stronger if additional real-world problem domains e.g. from natural language processing were presented, since the paper states that the method is applicable across diverse domains (line 87).


12. [$\star$]How does the proposed approach differ compared to this recent work [5] which very similarly proposes a cost-aware multi-step look ahead acquisition function? I think clarifying the difference with respect to this work is crucial in evaluating the novelty and contribution of the paper.



**Minor:**

- A very minor, however, for the sake of consistency in writing, either nonmyopic or "non-myopic" should be used (e.g. line 94).
- The repetition of the "equation" in lines 244 and 254 should be corrected.


> [1] Astudillo, R., Jiang, D., Balandat, M., Bakshy, E., & Frazier, P. (2021). Multi-step budgeted bayesian optimization with unknown evaluation costs. Advances in Neural Information Processing Systems, 34, 20197-20209.

> [2] Lee, E. H., Eriksson, D., Perrone, V., & Seeger, M. (2021, December). A nonmyopic approach to cost-constrained Bayesian optimization. In Uncertainty in Artificial Intelligence (pp. 568-577). PMLR.

> [3] Belakaria, S., Doppa, J. R., Fusi, N., & Sheth, R. (2023, April). Bayesian optimization over iterative learners with structured responses: A budget-aware planning approach. In International Conference on Artificial Intelligence and Statistics (pp. 9076-9093). PMLR.

> [4] Jian Wu and Peter Frazier. Practical two-step lookahead Bayesian optimization. Advances in neural
	information processing systems, 32, 2019

> [5] Liu, P., Wang, H., & Qiyu, W. (2023). Bayesian Optimization with Switching Cost: Regret Analysis and Lookahead Variants. Proceedings of the 32nd International Joint Conference on Artificial
Intelligence, IJCAI 2023. (pp. 4011-4018).

**Questions:**

1. How does the performance of the proposed method change against baselines under increased batch sizes?

2. How is the performance of the proposed approach against other nonmyopic BO methods given in the paper? (also see the weakness 1)

3. How do you evaluate the robustness of the proposed method with respect to the changes in reward function?

4. Is the proposed approach flexible and generalizable? Can it work under different acquisition functions?

5. Are the baselines other than the multistep tree implemented with a lookahead horizon=1 (myopic), is that right?

---

> ### Author Response · Authors · 2024-11-24
> **Responses for weaknesses**
>
> Dear Reviewer xEWL,
>
> Thank you for your valuable feedback. We would like to address the weaknesses and your questions below.
>
> Regarding your concerns about the quality of approximation of reward function (i.e. the surrogate model), we have included results from experiments with different numbers of initial points. Our findings indicate that with fewer initial points, the surrogate model struggles to accurately approximate the ground-truth function, which can lead to suboptimal optimization outcomes in all methods.
>
> [Figure for different inital samples](https://anonymous.4open.science/r/nonmyopia/images/initial_samples/NonmyopicBO%20-%20Initial_samples.png)
>
> Regarding your concerns about the flexibility and generalizability of our method, we would like to clarify them in the following. Our proposed method can indeed be seen as a generalized version of other myopic baselines used in our paper. Specifically, the $\ell(\cdot)$ function in line 188 can be replaced with any other acquisition function. In our paper, we adapted other acquisition functions such as Expected Improvement (EI), Probability of Improvement (PI), Simple Regret (SR), Upper Confidence Bound (UCB), and Knowledge Gradient (KG). Their optimization objectives of EI, PI, SR, and UCB are adapted as follows:
> \begin{equation}
> x = \arg \inf_{x \in \mathcal{X}} \left[ \mathbb{E} \_{p_t(f)} [\ell(f, x)] + \lambda c(x_{1:t}, x) \right],
> \end{equation}
> and that of KG is
> \begin{equation}
> x = \arg \inf_{x \in \mathcal{X}} \left[ \mathbb{E} \_{p_t(y | x)} [ \inf_{a \in \mathcal{A}} \{ \mathbb{E} \_{p \_{t+1}(f)} \ell(f, a)] + \lambda c(x_{1:t}, x, a) \} \right].
> \end{equation}
> Using these objective functions, we ensured that all methods are under the same conditions, thus demonstrating the effectiveness of our method in terms of lookahead capacity. The baseline results in our paper are already resulted	from the modified methods as presented above.
>
> Additionally, we have included experimental results on the ablation of the number of lookahead steps in the revised version. These results illustrate the relationship between the number of lookahead steps and the robustness of the optimization, providing insights into how the performance of our approach varies with different horizon lengths. Specifically, with a smaller lookahead horizon, the probability of being trapped by local optima increases, leading to suboptimal optimization in all nonmyopic methods.
>
> [Figure of ablation on lookahead steps](https://anonymous.4open.science/r/nonmyopia/images/lookahead/lookahead_0.05_r-spotlight_init-1_hidden64_rs64.png)
>
> Subsequently, regarding the effect of the quality of the surrogate model on the optimization process, we provide details about the number of training samples in Section 4.1 (line 349). We have included additional ablation results from experiments that examine the impact of different numbers of initial points (as we presented above) and different kernel choices (RBF and Matern with $\nu=1.5$). Our findings indicate that a poor prior with a bad surrogate model can lead to suboptimal solutions in all methods. We also observe that with any well-fitted kernels, the nonmyopic approach can achieve the global optimum, while the myopic approaches still struggle.
>
> [Figure of RBF kernel](https://anonymous.4open.science/r/nonmyopia/images/kernels/kernel_RBF_0.05_r-spotlight_init-1_hidden64_rs64.png)
> [Figure of Matern kernel](https://anonymous.4open.science/r/nonmyopia/images/kernels/kernel_Matern-1.5_0.05_r-spotlight_init-1_hidden64_rs64.png)
>
> In our experiment, we utilize a large language model (LLM) to perform the protein editing task. Theoretically, it is possible to fine-tune the LLM to edit all positions with all amino acid types. However, in practical scenarios [1, 2], protein editing is often constrained to a few positions with a limited number of amino acid types at a time. To reflect these practical constraints, we limited the number of positions and amino acids in our experiment. Moreover, constrained generation with an LLM is more challenging than allowing the LLM to change all positions of the sequence. We addressed this issue by carefully supervised fine-tuning the LLM before running the optimization process.
>
> [1] Kunkel, T. A. (1985). Rapid and efficient site-specific mutagenesis without phenotypic selection. Proceedings of the National Academy of Sciences, 82(2), 488–492.
> [2] Dalbadie-McFarland, G., Cohen, L. W., Riggs, A. D., Morin, M. J., Itakura, K., & Richards, J. H. (1982). Oligonucleotide-directed mutagenesis as a general and powerful method for studies of protein function. Proceedings of the National Academy of Sciences, 79(21), 6409–6413.

---

> ### Author Response · Authors · 2024-11-24
> **Responses for weaknesses (continue)**
>
> Regarding the ablation studies on the impact of different numbers of edits, we currently design the protein experiments to balance between demonstrating the effectiveness of our method and our computational budget. The number of edits required to reach the maximal reward protein is 12, so the number of BO steps must be at least 12. In practical scenarios, we will have a fixed, limited number of allowed steps. To make the experiment closer to a practical scenario, we assume that the maximal number of BO steps, ($T$), is 16. The results presented in our paper show that our method reaches the maximal protein at BO step 13, proving the effectiveness of our method. In short, setting the number of BO steps ($T$) larger than 16 is unnecessary in our case, and setting it smaller than 12 is inappropriate because of the problem setting.
>
> In practical protein design, there might be cases where we need to edit more positions or allow more amino acids. These requirements do not increase the computational budget because the neural network we chose is an LLM, which has a fixed cost for generating any proteins of the same length. Regarding the number of BO steps, if we need to increase it due to a larger search space, the complexity increases linearly. This is because, with the same number of lookahead steps, all BO steps have the same complexity. In short, our proposed method is scalable for larger search space even in a complex, semantic-rich environment.
>
> Nextly, we want to clarify our points about your concerns about BatchedBO, which is used in protein design. The idea of BatchedBO is sampling a batch of candidate protein simultaneously at each step using Thompson sampling. The difference between ours and BatchedBO is that we do not sample multiple $f$ instead we perform sampling when generating the sequences. For details, assuming the BatchedBO samples K $f$ at a BO step, then it computes the corresponding K generations. In our approach, we use LLM as the policy, thus the $f$ is the LLM’s parameters. Sampling multi $f$ in our case will be very costly and ineffective. Thus we perform sampling one $f$ and sampling when generation (using generation sampling techniques of LLM) and achieve the same number of K diverse generations. Doing this reduces the computational costs for generating candidate sequences.
>
> To further demonstrate our applicability, we have included additional results on protein design experiments using a different starting protein and a different synthetic function with 2 local modes. Furthermore, we are working on human travel optimization problems using maps captured from NASA (https://earthobservatory.nasa.gov/features/NightLights). These additions help to illustrate the versatility and robustness of our approach across different real-world problem domains.
>
>
> *Same starting protein, different $g(x)$ with 2 local modes*
>
> [Figure of distribution of fluorescence level](https://anonymous.4open.science/r/nonmyopia/images/additional_protein_exps/mutants_2p12_v1_v2_hop.png)\
> [Observed fluorescence level and Bayesian regret across BO steps](https://anonymous.4open.science/r/nonmyopia/images/additional_protein_exps/m1f2_yA_regret.png)
>
> *Different starting protein, same $g(x)$*
>
> [Figure of distribution of fluorescence level](https://anonymous.4open.science/r/nonmyopia/images/additional_protein_exps/mutants_2p12_v2_v1_hop.png)\
> [Observed fluorescence level and Bayesian regret across BO steps](https://anonymous.4open.science/r/nonmyopia/images/additional_protein_exps/m2f1_yA_regret.png)
>
> Regarding your suggested similar work [5], the primary difference between our approach and the one proposed in [5] lies in the integration of pathwise sampling and a variational network. While [5] introduces a cost-constrained lookahead acquisition function similar to ours, it does not incorporate pathwise sampling. Without pathwise sampling, the number of samples at each step grows exponentially. Additionally, without the variational network, the number of optimizing parameters also grows exponentially. By integrating pathwise sampling and a variational network, our method significantly reduces computational complexity, leading to an increased lookahead horizon and a higher chance of finding the global optimum. This allows us to optimize language models for more complex, semantic-rich tasks, addressing limitations that prior nonmyopic methods, including the one in [5], still face.
>
> * For minor representation mistakes: We thank the reviewer for pointing out the mistakes, and we have fixed them for the revision.

---

> ### Author Response · Authors · 2024-11-24
> **Responses for the questions**
>
> **Question 1:** How does the performance of the proposed method change against baselines under increased batch sizes?
>
> **Answer 1:** Thank you for your question. In the problem we are considering, at each BO step, the decision maker is allowed to make a single decision – which means the batch size is 1. Thus, all the methods we presented in our paper are designed to work with a batch size of 1.
> There is another term, batch size, used in the protein experiment in our paper, which might have caused some confusion. The term batch size in the protein design experiments refers to the batch size when fine-tuning the language model. This batch size is adjusted based on our computational budget and does not theoretically affect the proposed nonmyopic BO method.
>
>
> **Question 3:**  How do you evaluate the robustness of the proposed method with respect to the changes in reward function?
>
> **Answer 3:** We conducted an ablation study focusing on different initial points for training the surrogate model. Our findings indicate that with a smaller number of initial points, all methods (both myopic and nonmyopic) exhibit reduced performance due to insufficient information or signals about the mode in the optimization space.
>
> [Figure for different inital samples](https://anonymous.4open.science/r/nonmyopia/images/initial_samples/NonmyopicBO%20-%20Initial_samples.png)
>
>
> **Question 4:** Is the proposed approach flexible and generalizable? Can it work under different acquisition functions?
>
> **Answer 4:** Yes, our approach is both flexible and generalizable. We demonstrated its versatility by implementing it with various acquisition functions, including Expected Improvement (EI), Probability of Improvement (PI), Simple Regret (SR), Upper Confidence Bound (UCB), Knowledge Gradient (KG), and Multi-step Tree, as detailed in the paper. Specifically, we replace the $\ell(\cdot)$ function with other acquisition functions. Specifically, their optimization objectives of EI, PI, SR, and UCB are adapted to:
>
> \begin{equation}
> x = \arg \inf_{x \in \mathcal{X}} \left[ \mathbb{E} \_{p_t(f)} [\ell(f, x)] + \lambda c(x_{1:t}, x) \right],
> \end{equation}
>
> and that of KG is
> \begin{equation}
> x = \arg \inf_{x \in \mathcal{X}} \left[ \mathbb{E} \_{p_t(y | x)} [ \inf_{a \in \mathcal{A}} \{ \mathbb{E} \_{p \_{t+1}(f)} \ell(f, a)] + \lambda c(x_{1:t}, x, a) \} \right].
> \end{equation}
> Using these objective functions, we ensured that all methods are under the same conditions, thus demonstrating the effectiveness of our method in terms of lookahead capacity. The baseline results in our paper are already derived from the modified methods as presented above.
>
> **Question 5:** Are the baselines other than the multistep tree implemented with a lookahead horizon=1 (myopic), is that right?
>
> **Answer 5:** Other baselines (EI, PI, SR, UCB) are implemented as myopic approaches with a lookahead horizon equal to 0, which means these baselines optimize for immediate reward in the next step. The KG, which is equivalent to our approach without pathwise sampling and a variational network, is implemented with a lookahead horizon equal to 1. This means KG will look ahead for 1 step before making the final action. The mathematical optimization objectives of these baselines are presented as in the answer to the above question.

---

> ### Author Response · Authors · 2024-11-24
> **Response for weakness 1 regarding benchmarking the related works**
>
> Thank you for your feedback on the related works concerning non-myopic Bayesian optimization methods for dynamic cost settings. We appreciate the opportunity to clarify the relation between our work and the previous ones in terms of the cost structure. To facilitate the discussion, we survey the prior literature on the topic broadly and come up with a taxonomy based on two factors of the cost function: uncertainty and variability.
>
> - In terms of uncertainty, costs can be classified as known or unknown prior to making a decision. When the cost is unknown, it can be viewed as a random variable that can be modeled probabilistically.
>
> - In terms of variability, the cost structure can be categorized into dynamic costs, which vary based on the query history, and static costs, which remain fixed for a particular query over time. We note that when the cost is static, the BO community also studies two variations of this structure: heterogeneous cost and homogeneous cost. Homogeneous cost is the setting where the cost of all queries is the same, whereas heterogeneous cost is the setting where the cost of a query is a function of the query itself.
>
> Using this taxonomy, we classify the literature into four categories, as shown in the table below.
> |               | Known Cost                                                                                      | Unknown Cost                                                                                     |
> |---------------|-------------------------------------------------------------------------------------------------|--------------------------------------------------------------------------------------------------|
> | **Static Cost** | Do not vary based on the previous queries and predictable costs, easy to budget over time. Related literature: [4], [5], and [6]  | Do not vary based on the previous queries, but the actual amount is not fully known due to external factors.                                              Related literature: [1], [2], [3], and [7]   |
> | **Dynamic Cost** | Varies based on the previous queries, but can be quantified or predicted.                      Related literature: Ours | Varies based on the previous queries, and is difficult to predict precisely.                      Related literature: To the best of our knowledge, we have not yet encountered related work in this category, and we plan to work on this setting in our future direction. |
>
> To further illustrate the distinction between cost structures, we visualize the uncertainty and variability of these structures as probabilistic graphical diagrams below. In these diagrams, $f$ represents the target black-box function, $x$ denotes the input query, $y$ is the output value, and $c$ is the cost incurred by querying $x$. On the left — the dynamic-known cost structure — the cost of querying $x_3$ can depend on $x_1$ and $x_2$. On the right — the static-unknown cost structure — the cost of querying $x_3$ is independent of other queries.
>
> [Comparison of cost structures](https://anonymous.4open.science/r/nonmyopia/images/cost_structures.png)
>
>
> **References:**
>
> [1] Astudillo, R., Jiang, D., Balandat, M., Bakshy, E., & Frazier, P. (2021). Multi-step budgeted bayesian optimization with unknown evaluation costs. Advances in Neural Information Processing Systems, 34, 20197-20209.
>
> [2] Lee, E. H., Eriksson, D., Perrone, V., & Seeger, M. (2021, December). A nonmyopic approach to cost-constrained Bayesian optimization. In Uncertainty in Artificial Intelligence (pp. 568-577). PMLR.
>
> [3] Belakaria, S., Doppa, J. R., Fusi, N., & Sheth, R. (2023, April). Bayesian optimization over iterative learners with structured responses: A budget-aware planning approach. In International Conference on Artificial Intelligence and Statistics (pp. 9076-9093). PMLR.
>
> [4] Jian Wu and Peter Frazier. Practical two-step lookahead Bayesian optimization. Advances in neural information processing systems, 32, 2019
>
> [5] Nyikosa, F. M., Osborne, M. A., & Roberts, S. J. (2018). Bayesian optimization for dynamic problems. arXiv preprint arXiv:1803.03432.
>
> [6] Lam, R., Willcox, K., & Wolpert, D. H. (2016). Bayesian optimization with a finite budget: An approximate dynamic programming approach. Advances in Neural Information Processing Systems, 29.
>
> [7] Luong, P., Nguyen, D., Gupta, S., Rana, S., & Venkatesh, S. (2021). Adaptive cost-aware Bayesian optimization. Knowledge-Based Systems, 232, 107481.

---

> ### Author Response · Authors · 2024-11-24
> **Response for weakness 1 regarding benchmarking the related works (continue)**
>
> Our problem setting focuses on optimizing within a *known and dynamic* cost setting, which is an important cost structure in many practical applications as we motivated earlier. Previous literature has developed methods for complementary-but-distinct cost settings, and we believe that those methods are not suitable for the setting studied in this paper. We elaborate on our position further below by analyzing the cost structure of the suggested papers:
> - The papers *“Multi-step budgeted Bayesian optimization with unknown evaluation costs”*, *“A nonmyopic approach to cost-constrained Bayesian optimization”*, and *“Adaptive cost-aware Bayesian optimization”* address cost structures characterized by unknown, heterogeneous costs. For instance, in the hyperparameter optimization (HPO) problem studied in these papers, the cost of evaluating a hyperparameter set (i.e., training the target model with that set) is unknown but static for a particular model and a set of hyperparameters and does not depend on previously chosen sets.
> - The paper *“Bayesian optimization over iterative learners with structured responses: A budget-aware planning approach”* also incorporates an unknown, heterogeneous cost structure. The key distinction from the previously mentioned works is the inclusion of an additional cost factor: the number of training epochs for evaluating a hyperparameter set. In this context, for a given location \( x \) (a set of hyperparameters) and a specific number of training epochs, the cost is unknown but fixed, as it does not depend on prior queries (previously chosen sets of hyperparameters) or prior choices of training epochs.
>
> - Finally, the papers *“Practical two-step lookahead Bayesian optimization”*, *“Bayesian optimization for dynamic problems”*, and *“Bayesian optimization with a finite budget: An approximate dynamic programming approach”* focus on Bayesian optimization without accounting for cost structures. These works implicitly assume that all locations in the search space have the same, constant, and known cost. This implies that the cost is either zero or any fixed constant and, therefore, not subject to optimization.
>
> Based on the above evidence, the settings in these works are fundamentally different from ours.
>
> Regarding the non-myopic methods presented in the suggested papers [1–4], these approaches extend the Expected Improvement acquisition function to address non-myopic optimization challenges. While they provide valuable insights, these methods directly optimize free variables in a multi-step tree (MST), which introduces an exponential increase in the number of optimization variables as the lookahead horizon grows. In our context, incorporating their lookahead mechanisms would amount to combining a multi-step tree structure with a dynamic cost function, an approach that is already benchmarked in our work. The optimization objectives of these lookahead methods, once adapted, align with Equation (1) (lines 188–190) in our paper. Our planning algorithm distinguishes itself from the literature by integrating a policy neural network. In the context of this paper on the dynamic and known cost setting, we believe that our experiments have sufficiently benchmarked the previous planning strategy presented in the suggested papers. We are eager to hear the reviewer's thoughts on this to further improve our work and welcome any specific suggestions from the reviewer on how to incorporate the related nonmyopic BO methods as additional baselines for our setting. We again thank you for your thoughtful questions and we look forward to hearing more from you.
>
> **References:**
> [1] Astudillo, R., Jiang, D., Balandat, M., Bakshy, E., & Frazier, P. (2021). Multi-step budgeted bayesian optimization with unknown evaluation costs. Advances in Neural Information Processing Systems, 34, 20197-20209.
>
> [2] Lee, E. H., Eriksson, D., Perrone, V., & Seeger, M. (2021, December). A nonmyopic approach to cost-constrained Bayesian optimization. In Uncertainty in Artificial Intelligence (pp. 568-577). PMLR.
>
> [3] Belakaria, S., Doppa, J. R., Fusi, N., & Sheth, R. (2023, April). Bayesian optimization over iterative learners with structured responses: A budget-aware planning approach. In International Conference on Artificial Intelligence and Statistics (pp. 9076-9093). PMLR.
>
> [4] Jian Wu and Peter Frazier. Practical two-step lookahead Bayesian optimization. Advances in neural information processing systems, 32, 2019
>
> [5] Nyikosa, F. M., Osborne, M. A., & Roberts, S. J. (2018). Bayesian optimization for dynamic problems. arXiv preprint arXiv:1803.03432.
>
> [6] Lam, R., Willcox, K., & Wolpert, D. H. (2016). Bayesian optimization with a finite budget: An approximate dynamic programming approach. Advances in Neural Information Processing Systems, 29.
>
> [7] Luong, P., Nguyen, D., Gupta, S., Rana, S., & Venkatesh, S. (2021). Adaptive cost-aware Bayesian optimization. Knowledge-Based Systems, 232, 107481.

---

> ### Author Response · Authors · 2024-11-24
> **Response for weakness 1 regarding the computational efficiency**
>
> Regarding computational efficiency, we demonstrate theoretical improvements. Specifically, the use of a policy network reduces the complexity to $O(1)$ in terms of the number of optimizing parameters. In contrast, previous nonmyopic BO methods exhibit a complexity of $O(k^T)$, where $k$ is the number of sampling samples per BO lookahead step and $T$ is the number of lookahead steps (lines 232-237).
>
> Additionally, Pathwise sampling further enhances efficiency. The complexity is reduced to $O(C+K)$, where $C$ is the number of queried points and $K$ is a constant. This marks a significant improvement over traditional Monte Carlo sampling methods, which have a complexity of $O(K m^3)$, with $m$ representing the number of inducing points.
>
> You can find more about our implementation at his anonymized Github: https://anonymous.4open.science/r/nonmyopia

---

> > ### Comment · Reviewer_xEWL · 2024-11-27
> > **Thank you for your response; I have still concern**
> >
> > Thanks a lot to the authors, with the addition of ablation studies and discussions. I have the following additional comments about your response:
> >
> > - **[Weakness 12]:** My concern stated in bullet 12 (which I labeled as [$\star$] to raise attention) remains underaddressed. As I have stated previously, it is **crucial** to evaluate this paper's novelty and contribution compared to the related work [1], which also proposes a nonmyopic BO approach. Since it is the SOTA method that directly addresses the same problem, I think the paper should have compared the proposed method against it. It is the most appropriate baseline to consider than using modified versions of myopic approaches.
> > - **[Limited empirical evaluation]:** I still have the concern that even though the benchmarked related works (in your response) are distinguished regarding different types of cost structures, they are still nonmyopic, and/or cost-constrained approaches, which are the closest methods to compare against, than modified versions of myopic traditional BO methods. In my opinion, empirical evaluation against myopic approaches (which, also assume static costs) supports the proposed approach's effectiveness even less than comparing against more state-of-the-art methods.
> > - I also think that the provided results and discussions should have been directly presented in the updated manuscript, which would have enriched the content.
> > - **[Question on fewer initial points result]:** It seems that MSL is performing better than the proposed approach, particularly under a very restricted number of initial points. What do you think about this result; could you provide more insight about the robustness of your approach compared to MSL?
> >
> > Overall, although not included in your response, I appreciate the ongoing effort to address applicability of the proposed method under different problem domains. I believe adding the reported additional ablations to the paper would definitely improve the analysis. However, considering that key analyses are limited, I tend to keep my current score.
> >
> > [1] Liu, P., Wang, H., & Qiyu, W. (2023). Bayesian Optimization with Switching Cost: Regret Analysis and Lookahead Variants. Proceedings of the 32nd International Joint Conference on Artificial Intelligence, IJCAI 2023. (pp. 4011-4018)

---

> > > ### Author Response · Authors · 2024-12-01
> > > **Response for the weekness 12 and empirical evaluation**
> > >
> > > Dear Reviewer xEWL,
> > >
> > > We would like to thank you for your thoughtful feedback, which helped us improve our paper. Regarding Weakness 12, we have addressed this at the end of the [second response](https://openreview.net/forum?id=IiAckbuccF&noteId=qZpoRPkbjf). We apologize for not making it clearer. Below, we would like to justify our approach compared to the paper [1] you mentioned in more detail.
> > >
> > > [1] introduce a general switching cost of queries, implementing the Euclidean cost (Section 2.2, [1]). The lookahead proposed in [1] is equivalent to cost-constraint MSL (multi-step tree), which we have already studied as a baseline in our paper. To the best of our knowledge, MSL is the state-of-the-art nonmyopic Bayesian optimization method.
> > >
> > > Regarding the discussion we have posted, we have incorporated the new results and suggestions from the reviewer into the rebuttal version of this paper. We again thank the reviewer for their insightful comments. We hope this addresses your concerns and we looking for your continued feedback.
> > >
> > > [1] Liu, P., Wang, H., & Qiyu, W. (2023). Bayesian Optimization with Switching Cost: Regret Analysis and Lookahead Variants. Proceedings of the 32nd International Joint Conference on Artificial Intelligence, IJCAI 2023. (pp. 4011-4018)

---

> > > ### Author Response · Authors · 2024-12-02
> > > **Response for the question on fewer initial points results**
> > >
> > > Dear Reviewer xEWL,
> > >
> > > Regarding the performance of MSL and our method with a small number of initial samples, as we mentioned before, the difference between the MSL implementation in our paper and our method lies in the variational network. The primary role of the variational network is to reduce the number of optimization parameters in more complex, semantic-rich environments such as protein design. In simpler environments like 2D synthetic functions, optimizing directly on decision variables (as in MSL) can potentially lead to better performance in some cases because the amortized network originally focuses on exploitation.
> > >
> > > To address this issue, we increase the concentration parameter of the von Mises–Fisher distribution to sample output noise of the amortized network. We have included new results from the ablation study where we increased the concentration parameter in the figure below. Now, we can observe that the performance of our method is comparable to MSL.
> > >
> > > [Figure for initial sample ablation study](https://anonymous.4open.science/r/nonmyopia/images/initial_samples/NonmyopicBO%20-%20Initial_samples%20-%20Improved.png)

---

### Author Response · Authors · 2024-12-01
**General Response**

Dear reviewers and chairs,

We deeply appreciate the reviewers for their detailed and insightful feedback!
In response to the reviewers' feedback, we have made substantial improvements to our paper by addressing critical areas highlighted for revision.

First, we conducted a comprehensive survey on cost structures and categorized the literature into four types based on uncertainty and variability. Specifically, we classified them into known-unknown costs and dynamic-static costs, as shown in the table below. In this taxonomy, the related works [1-7] exhibit different cost structures, making them unsuitable for direct inclusion as baselines. Fortunately, the lookahead mechanisms proposed in [1-4] are equivalent to MSL, which is already benchmarked as a baseline in our paper.

|               | Known Cost                                                                                      | Unknown Cost                                                                                     |
|---------------|-------------------------------------------------------------------------------------------------|--------------------------------------------------------------------------------------------------|
| **Static Cost** | Do not vary based on the previous queries and predictable costs, easy to budget over time. Related literature: [4], [5], and [6]  | Do not vary based on the previous queries, but the actual amount is not fully known due to external factors.                                              Related literature: [1], [2], [3], and [7]   |
| **Dynamic Cost** | Varies based on the previous queries, but can be quantified or predicted.                      Related literature: Ours | Varies based on the previous queries, and is difficult to predict precisely.                      Related literature: To the best of our knowledge, we have not yet encountered related work in this category, and we plan to work on this setting in our future direction. |

Regarding [8], it introduces a general switching cost of queries, implementing the Euclidean cost (Section 2.2, [8]). The lookahead proposed in [8] is equivalent to cost-constraint MSL (multi-step tree), which we have already studied as a baseline in our paper. To the best of our knowledge, MSL is the state-of-the-art nonmyopic Bayesian optimization method. The primary difference between our approach and the one proposed in [8] lies in the integration of pathwise sampling and a variational network. While [8] introduces a cost-constrained lookahead acquisition function similar to ours, it does not incorporate pathwise sampling or a variational network. Without pathwise sampling, the number of samples at each step grows exponentially. Additionally, without the variational network, the number of optimizing parameters also grows exponentially. By integrating pathwise sampling and a variational network, our method significantly reduces computational complexity, leading to an increased lookahead horizon and a higher likelihood of finding the global optimum. This allows us to optimize language models for more complex, semantic-rich tasks, addressing limitations that prior nonmyopic methods, including the one in [8], still face.


**References:**

[1] Astudillo, R., Jiang, D., Balandat, M., Bakshy, E., & Frazier, P. (2021). Multi-step budgeted bayesian optimization with unknown evaluation costs. Advances in Neural Information Processing Systems, 34, 20197-20209.

[2] Lee, E. H., Eriksson, D., Perrone, V., & Seeger, M. (2021, December). A nonmyopic approach to cost-constrained Bayesian optimization. In Uncertainty in Artificial Intelligence (pp. 568-577). PMLR.

[3] Belakaria, S., Doppa, J. R., Fusi, N., & Sheth, R. (2023, April). Bayesian optimization over iterative learners with structured responses: A budget-aware planning approach. In International Conference on Artificial Intelligence and Statistics (pp. 9076-9093). PMLR.
(Varying by the number of BO steps)

[4] Jian Wu and Peter Frazier. Practical two-step lookahead Bayesian optimization. Advances in neural information processing systems, 32, 2019

[5] Nyikosa, F. M., Osborne, M. A., & Roberts, S. J. (2018). Bayesian optimization for dynamic problems. arXiv preprint arXiv:1803.03432.

[6] Lam, R., Willcox, K., & Wolpert, D. H. (2016). Bayesian optimization with a finite budget: An approximate dynamic programming approach. Advances in Neural Information Processing Systems, 29.

[7] Luong, P., Nguyen, D., Gupta, S., Rana, S., & Venkatesh, S. (2021). Adaptive cost-aware Bayesian optimization. Knowledge-Based Systems, 232, 107481.

[8] Liu, P., Wang, H., & Qiyu, W. (2023). Bayesian Optimization with Switching Cost: Regret Analysis and Lookahead Variants. Proceedings of the 32nd International Joint Conference on Artificial Intelligence, IJCAI 2023. (pp. 4011-4018).

---

### Author Response · Authors · 2024-12-01
**General Response (continue)**

Secondly, to illustrate the advantages of non-myopic methods, we created a visualization as shown in the figure below. The target is to optimize a 1D function using a myopic approach (left) and our non-myopic method (right) with a 6-step lookahead. The top row highlights the receptive field—the region the decision-maker can see prior to making a decision—which varies with the planning horizon. In the myopic setting (left), the receptive field is narrow, focusing on short-term gains, as indicated by the limited shaded region. In contrast, the non-myopic approach (right) extends the receptive field, enabling the method to account for longer-term outcomes.

[Figure for demonstrating differences between myopic and nonmyopic method](https://anonymous.4open.science/r/nonmyopia/images/NonmyopicBO%20-%20MainFig.png)

Both methods operate within the same radius (depicted as the area between the two dashed lines in the top row), but the key difference lies in the non-myopic method’s broader receptive field. This expanded perspective allows the method to better anticipate and account for future rewards. In contrast, the myopic approach, while potentially equipped with prior knowledge of the global optimum, prioritizes immediate rewards. This short-term focus may prevent it from overcoming local valleys to reach the global optimum.

Thirdly, we conducted comprehensive ablation studies to assess the robustness and flexibility of our approach under various settings, including different lookahead steps, initial training set sizes, and kernel choices. These studies demonstrate our method’s consistent performance across diverse configurations and underscore its adaptability beyond traditional myopic techniques. To further validate scalability and practical relevance, we expanded the experimental section to include real-world applications such as optimizing human travel paths using map data from NASA’s Earth Observatory and additional protein design tasks with a more challenging synthetic function. These additions illustrate that our method is capable of scaling effectively across distinct domains.

In terms of implementation details, we elaborated on our hyperparameter selection, balanced effectiveness with computational budget considerations, and conducted additional sensitivity analyses for UCB baselines to provide a transparent evaluation. We also addressed minor writing inconsistencies and presentation mistakes. Finally, we provided detailed explanations on integrating Llama-3.2 3B as the policy network, describing our strategies for handling failure cases when constraints are unmet, thus ensuring robust optimization. With these revisions, we demonstrate the method’s versatility and robustness across a range of challenging scenarios, thereby thoroughly addressing reviewer concerns.

Our source code is available at this anonymized Github: https://anonymous.4open.science/r/nonmyopia

---

### Meta-Review · Area_Chair_PDRL · 2024-12-21

**Metareview:**

This paper investigated BO in dynamic cost settings, which has practical relevance for real-world applications. The authors propose a novel approach using neural network policies and pathwise sampling to perform nonmyopic BO efficiently, reducing the computational complexity of multi-step planning. The method is evaluated on synthetic functions, a protein design task, and a (newly added) human travel optimization task.

**Strengths:**
- The problem studied is important and practically relevant
- The proposed method introduces a novel combination of neural network policies and pathwise sampling to handle dynamic cost settings.
- The authors expanded the experimental section with additional tasks and conducted thorough ablation studies on hyperparameters and kernel choices.

**Weaknesses:**
- The paper does not provide a direct empirical comparison to Liu et al. (2023), a state-of-the-art nonmyopic BO method addressing a similar problem, despite claims that the MSL baseline captures its essence.
- The explanation of the "receptive field" for nonmyopic vs. myopic methods is considered insufficiently rigorous, with one reviewer requesting stronger theoretical analysis.

While the problem is important and the proposed method has merit, the lack of direct comparison to a key relevant baseline and the concerns about theoretical rigor and empirical scope weigh heavily. Despite the authors' significant efforts in the rebuttal, these issues remain unresolved for the majority of reviewers. We encourage the authors to address these concerns in a future submission, as the proposed approach has potential for impactful contributions to the field.

**Additional Comments On Reviewer Discussion:**

The reviewers appreciated the authors' efforts during the rebuttal phase, including the expanded experiments and clarifications. However, a consensus emerged that the lack of a direct empirical comparison to Liu et al. (2023), a key nonmyopic BO baseline, remains a critical weakness. Reviewer xEWL elaborated that Liu et al.’s method, with its path-dependent rollout approximation, offers a stronger foundation for longer horizons compared to the MSL implementation described in the paper.

Overall, the reviewers acknowledge the importance of the problem and the authors' substantial efforts but feel that further improvements in empirical and theoretical aspects are necessary for publication.

---

### Decision · Program_Chairs · 2025-01-22

Reject